# Large Language Models as Biomedical Hypothesis Generators: A Comprehensive Evaluation

**Biqing Qi** [1,4] *  **Kaiyan Zhang** [1,2] *  **Kai Tian** [1,2]  **Haoxiang Li** [1]  **Zhang-Ren Chen** [6]
**Sihang Zeng** [5]  **Ermo Hua** [1,2]  **Jin-Fang Hu** [6] †  **Bowen Zhou** [1,3] †

[1] Tsinghua University   [2] Frontis.AI   [3] Shanghai AI Laboratory
[4] Harbin Institute of Technology   [5] University of Washington
[6] The First Affiliated Hospital of Nanchang University

qibiqing7@gmail.com , zhang-ky22@mails.tsinghua.edu.cn, zhoubowen@tsinghua.edu.cn

## Abstract

The rapid growth of biomedical knowledge has outpaced our ability to efficiently extract insights and generate novel hypotheses. Large language models (LLMs) have emerged as a promising tool to revolutionize knowledge interaction and potentially accelerate biomedical discovery. In this paper, we present a comprehensive evaluation of LLMs as biomedical hypothesis generators. We construct a dataset of background-hypothesis pairs from biomedical literature, carefully partitioned into training, seen, and unseen test sets based on publication date to mitigate data contamination. Using this dataset, we assess the hypothesis generation capabilities of top-tier instructed models in zero-shot, few-shot, and fine-tuning settings. To enhance the exploration of uncertainty, a crucial aspect of scientific discovery, we incorporate tool use and multi-agent interactions in our evaluation framework. Furthermore, we propose four novel metrics grounded in extensive literature review to evaluate the quality of generated hypotheses, considering both LLM-based and human assessments. Our experiments yield two key findings: 1) LLMs can generate novel and validated hypotheses, even when tested on literature unseen during training, and 2) Increasing uncertainty through multi-agent interactions and tool use can facilitate diverse candidate generation and improve zero-shot hypothesis generation performance. However, we also observe that the integration of additional knowledge through few-shot learning and tool use may not always lead to performance gains, highlighting the need for careful consideration of the type and scope of external knowledge incorporated. These findings underscore the potential of LLMs as powerful aids in biomedical hypothesis generation and provide valuable insights to guide further research in this area. Data and code at https://github.com/TsinghuaC3I/LLM4BioHypoGen/.

## 1 Introduction

Biomedical research is a driving force behind advancements in human health and well-being. However, the exponential growth of biomedical data and literature has made it increasingly difficult for researchers to keep pace with the latest discoveries and generate novel hypotheses. Large language models (LLMs) (Touvron et al., 2023a;b; Jiang et al., 2024) have emerged as a promising solution to this challenge, offering the potential to revolutionize the way we interact with and discover knowledge in the biomedical domain(Wang et al., 2023a; Taylor et al., 2022). At the core of the knowledge discovery process is the formulation of sound hypotheses (Zhong et al., 2023; Boiko et al., 2023). Currently, both ChatGPT, GPT-4,

---

* Equal contributions.
† Corresponding authors.

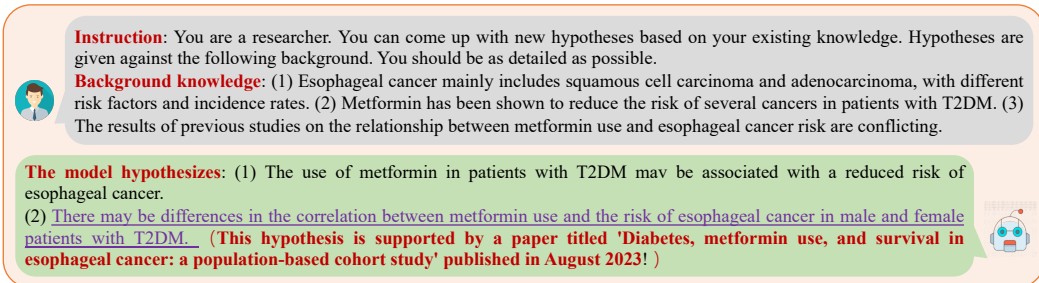

Figure 1: This illustration demonstrates a generated hypothesis using the fine-tuned 65B LLaMA model within our specially constructed dataset. The generated hypothesis closely aligns with the findings in existing literature published subsequent to the training sets.

and other open-source LLMs undergo extensive pre-training on a substantial amount of data. The analysis and validation of related hypothesis generation work rely on these models (Wang et al., 2023b; Yang et al., 2023b). However, due to the non-traceability of the training data, these endeavors cannot ensure data invisibility, i.e., the inability to guarantee non-overlapping between test and training data. This limitation hinders the assessment and analysis of knowledge discovery in LLMs under zero-shot conditions.

In this paper, we present a comprehensive evaluation of LLMs as biomedical hypothesis generators. We construct a dataset of background-hypothesis pairs from biomedical literature, carefully partitioned into training, seen, and unseen test sets based on publication date to mitigate data contamination. Using this dataset, we assess the hypothesis generation capabilities of top-tier instructed models in zero-shot, few-shot, and fine-tuning settings. To enhance the exploration of uncertainty, a crucial aspect of scientific discovery, we incorporate tool use (Schick et al., 2023a) and multi-agent interactions (Xi et al., 2023b) in our evaluation framework. These strategies are inspired by uncertainty exploration in reinforcement learning (Watkins & Dayan, 1992; Schulman et al., 2017) and aim to examine the influence of uncertainty on knowledge discovery and the primary factors contributing to zero-shot capability.

Our experiments yield two key findings: 1) LLMs can generate novel and validated hypotheses, even when tested on literature unseen during training (Figure 1), and 2) Increasing uncertainty through multi-agent interactions and tool use can facilitate diverse candidate generation and improve zero-shot hypothesis generation performance. However, we also observe that the integration of additional knowledge through few-shot learning and tool use may not always lead to performance gains, highlighting the need for careful consideration of the type and scope of external knowledge incorporated.

These findings underscore the potential of LLMs as powerful aids in biomedical hypothesis generation and provide valuable insights to guide further research in this area. Our specific contributions include:

- We pioneer the rigorous validation of LLMs in zero-shot and few-shot hypothesis generation through the creation of temporal biomedical instruction data and innovative experiments for in-depth analysis and evaluation.

- Our findings reveal that LLMs exhibit foundational higher-order reasoning abilities and can generate novel hypotheses, offering fresh empirical insights for knowledge discovery.

- We develop multidimensional metrics for evaluating hypotheses with GPT-4 and human. The correlation between these evaluations suggests LLMs' substantial role in hypothesis evaluation.

- We propose a LLM-based multi-agent framework for hypothesis generation. This system facilitates collaborative analysis among various roles and tools, enhancing our understanding of the influenced factors of LLM-based proposers.

## 2 Preliminary

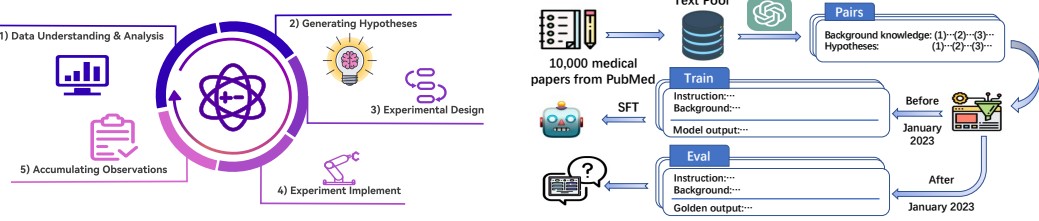

(a) Scientific Discovery Process      (b) Dataset Construction Pipeline

Figure 2: (a) The iterative loop of scientific discovery involves a cyclical process: observations and data from previous experiments are analyzed, leading to the generation of new hypotheses. These hypotheses then guide the design of subsequent experiments, producing fresh data to perpetuate the cycle. (b) We execute the automated data partitioning pipeline, using publication dates as the basis, to mitigate the risk of data contamination.

In this section, we delineate the problem of hypothesis generation. Subsequently, we elucidate the process for developing biomedical-specific datasets. We then analyze the constructed datasets, examining aspects of data contamination and the semantic distribution across different data settings.

### 2.1 Problem Definition

As shown in Figure 2a, hypothesis generation typically transpires following an in-depth analysis of literature and detailed examination of specific phenomena, playing a pivotal role in the scientific discovery process. In order to improve the assessment of the hypothesis proposition capabilities of LLMs, we formalize this process as a text completion task. Given dataset $D$, an instruction $I$, and text pairs $(X_i, Y_i)_{i=1}^{n}$ containing background knowledge and corresponding hypotheses extracted from medical papers, our objective is to assess model $M$ by having it generate hypotheses based on the task instruction and background knowledge, i.e., $M(I, X_i) = Y_i$, for each $i \in 1, ..., n$. The objective function is formulated as:

$$y^* = \arg \max_{y_1, ..., y_n} \prod_{t=1}^{n} P(y_t | y_1, ..., y_{t-1}, I, X).$$

### 2.2 Dataset Construction

Existing LLMs, including ChatGPT and Llama, encounter challenges in retracing their training data, complicating efforts to ensure non-overlap with the test set. To rigorously assess the hypothesis generation capability of LLMs, we initially create evaluation datasets based on literature, taking into account the publication date.

Unlike previous studies that assess data contamination through the correlation between question entropy and accuracy (Hendrycks et al., 2020; Wei et al., 2023b; Zhou et al., 2023), our method directly controls visibility by considering both the training date and the publication date. This method ensures that the test data remains non-visible, addressing a limitation overlooked in earlier methods. As depicted in Figure 2b, the year 2023 has been established as the cut-off point, coinciding with the publication date of the majority of top-tier LLMs. The training dataset includes literature published before January 2023, while the test dataset contains literature after January 2023. This arrangement creates pairs of data with background knowledge and corresponding hypothesis proposals. We strictly follow the standard pipeline as outlined in Self-Instruct (Wang et al., 2022) for our data generation process: 1) Compose the paper set based on the topic and content of the literature. 2) Utilize ChatGPT and GPT-4 to summarize the literature knowledge. 3) Generate background knowledge-assume pairs. 4) Filter low-quality data by publishers. 5) Split the dataset according to publication time.

While this approach allows for efficient dataset creation, we acknowledge that the accuracy of ChatGPT's extractions and any biases introduced in the process warrant further evaluation. Manual annotation of a subset of the data could help assess the quality and reliability of the automated extraction.

Adhering to the above pipeline outlined, we ultimately acquired two distinct types of datasets. 1) **Seen dataset** This dataset comprises 2700 background and hypothesis pairs sourced from literature published before January 2023. This dataset was partitioned into training (2500) and validation (200) subsets (as well as seen test set). It is consistent with the corpus that the LLMs have been exposed to. 2) **Unseen dataset** The unseen dataset consists of 200 pairs extracted from papers published in August 2023, which the LLMs have not encountered during training and are used for testing purposes.

More detailed information with respect to publication data and topic distributions of constructed dataset can be seen in Appendix C.1.

## 3 Can LLMs Truly Generate Zero-Shot Hypotheses?

In this section, we conduct a thorough evaluation of premier LLMs across a range of experimental settings to determine their ability to effectively generate hypotheses and analyze influenced factors.

### 3.1 Experiment Setup

Initially, we present the models being evaluated, outline the experimental settings, and describe the metrics used for evaluation.

**Models** For a fair comparison, we exclusively evaluate LLMs trained on corpora before March 2023 to avoid data contamination. We consider three categories of models: 1) API-based LLMs: this is mainly ChatGPT. 2) General domain instructed LLMs: These models consist of open-source models that have undergone fine-tuning based on Llama using general domain instructions. We primarily choose the top-tier models based on rankings before Setember 2023 in the Alpaca Eval Leaderboard [1]. 3) Specific domain instructed LLMs: These include PMC-LLaMA (Wu et al., 2023), and MedAlpaca (Han et al., 2023), which are trained on a variety of sources in medicine domain. We summarize the training data and publication dates for each models in Appendix C.2.

**Prompts and Finetuning** To ensure a consistent output format across different models, we create prompts in two formats: zero-shot and few-shot examples. We adopt a 5-shot format, selecting examples from the training set before January 2023 using both randomly sampled and similarity retrieval methods. To assess the hypothesis generation capability beyond zero-shot, we identify the top-performing open-source models through prompting-based evaluation. Finally, we proceed to fine-tune `WizardLM-13B-V1.2` with the background and hypothesis pairs for further comparison. We provide more details in Appendix F.

**Evaluation Metrics** Given the inherent uncertainty beyond established ground truth in hypothesis generation, we undertake evaluations with and without predefined golden hypotheses. 1) **With ground truth:** In evaluations with golden hypotheses, we employ standard text generation metrics, including BLEU (Papineni et al., 2002) and ROUGE (Lin, 2004) , to assess word overlap between the generated outputs and the ground truth. 2) **Without ground truth:** Considering the vastness of the hypothesis space, renders it difficult to comprehensively assess the quality of generated hypotheses using word overlap metrics alone. To provide a more comprehensive evaluation of the generated hypotheses from multiple facets, we devise four metrics for evaluation (Zhong et al., 2023):

- *Novelty*: Does the hypothesis introduce new information or perspectives?
- *Relevance*: How closely is the hypothesis related to the topic or question?

---

[1]https://tatsu-lab.github.io/alpaca_eval/

- *Significance*: What impact does the hypothesis have on understanding or addressing the problem?
- *Verifiability*: Can the hypothesis be tested using existing methods or data?

Furthermore, inspired by recent research that highlights LLMs as proficient annotators (Gilardi et al., 2023; Liu et al., 2023), demonstrating a strong correlation with human ratings, we employ GPT-4 for automated evaluation, where both the generated hypotheses and the provided background are evaluated across these aspects. The scoring scale ranges from 0 to 3, where a higher score indicates superior results. Additionally, we solicit GPT-4 to furnish a step-by-step explanation to substantiate the assigned score. We also conduct human evaluation for the top-tier models identified in the evaluation of GPT-4 in Section 3.4.

## 3.2 Experiment Results

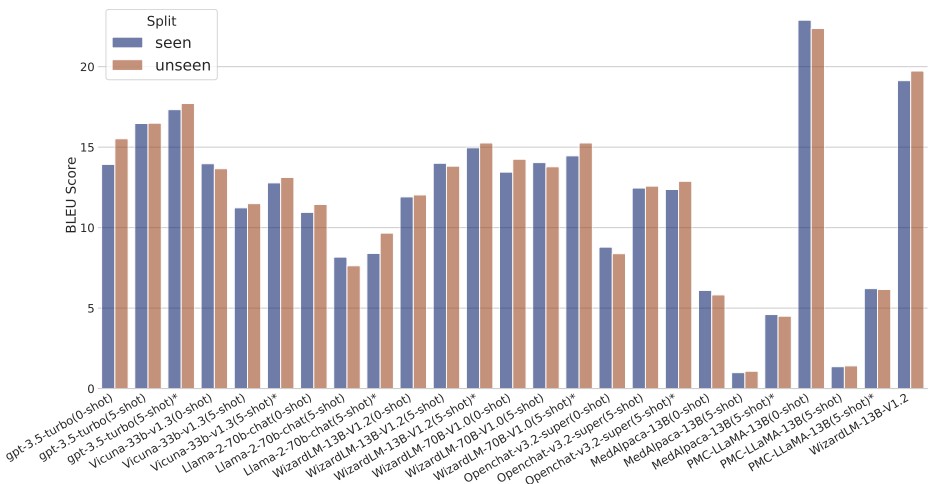

Figure 3: This figure displays the BLEU scores on both seen and unseen datasets.

This section presents the results of hypothesis generation across various models in both zero-shot and few-shot settings. We primarily analyze the results from two perspectives: the impact of the zero-shot setting and the influence of introducing external knowledge on hypothesis generation.

### 3.2.1 Results of Zero-shot Setting

The results presented in Figure 3 and 4 demonstrate the significant impact of zero-shot settings in improving hypothesis generation, particularly in terms of fostering high novelty. Detailed results are provided in Appendix D.1. In this section, we primarily analyze these results from two critical perspectives as outlined below:

**Zero-shot vs. Few-shot** Figure 4 demonstrates that nearly all models, particularly `WizardLM` series models, and `Openchat-v3.2-super`, show enhanced novelty capabilities in a zero-shot setting. Concurrently, these models also demonstrate superior verifiability when presented with few-shot literature examples. This indicates a trade-off in the hypothesis generation capacity of LLMs, necessitating careful consideration of specific constraints. Notably, there are divergent results concerning medical domain adaptation in LLMs, which will be further discussed in the subsequent Section 3.2.2.

**Seen test-set vs. Unseen test-set** Despite the inclusion of literature published before 2023 in the pre-training corpus of most LLMs, we have delineated "seen" and "unseen" test sets, with the "unseen" test set being considered for zero-shot analysis. Generally, LLMs are likely to show enhanced performance on the "seen" test set, potentially due to the memorization of knowledge acquired during training, resulting in a superior performance

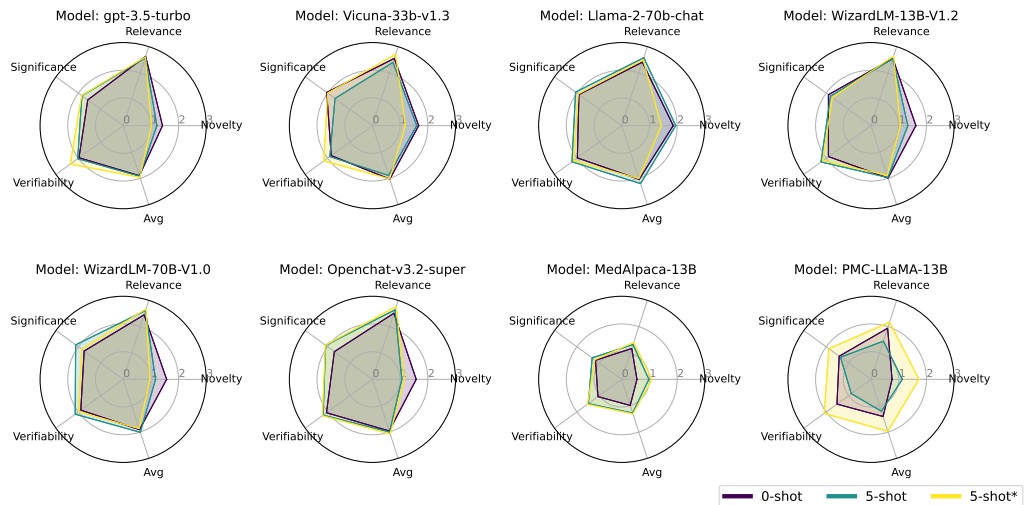

Figure 4: This figure depicts a comparative analysis of multiple models across distinct prompting paradigms, such as zero-shot, sampled and similarity retrieval-based few-shot.

compared to the "unseen" test set. Contrarily, our findings presented in Figure 3 reveal that LLMs tend to exhibit better performance on the "unseen" test set. We hypothesize that the intricacies involved in hypothesis generation may impede LLMs' ability to effectively utilize the parameterized "dark knowledge".

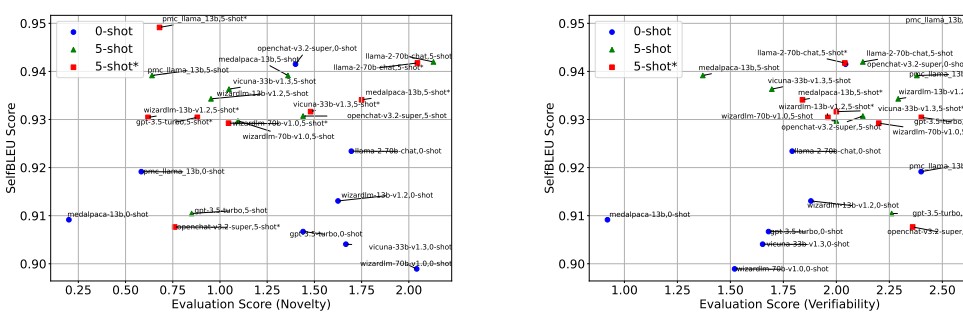

(a) Correlation of uncertainty and novelty.    (b) Correlation of uncertainty and verifiability.

Figure 5: This figure elucidates the correlation between uncertainty and evaluation scores for all models, encompassing both zero-shot and few-shot settings, and incorporating both sampled and similarity retrieval few-shot prompts.

### 3.2.2 Results of External Knowledge

We further extend analysis to the influence of external knowledge, including few-shot examples, domain adaptation, and instruction tuning. The findings indicate that these factors do not consistently enhance performance across all metrics.

**Few-Shot Examples Enhance Verifiability but Decrease Novelty.** Regarding word overlap metrics, including BLEU and ROUGE, most models, especially `gpt-3.5-turbo` and `WizardLM` series models, show improved performance when provided with in-context examples as shown in Figure 3. However, it's important to note that these few-shot prompts significantly increase verifiability while simultaneously leading to lower novelty compared to zero-shot results as shown in Figure 4.

| Category | Model | ChatGPT Eval.Avg | Human Eval | | | | | Avg Coefficient | |
|---|---|---|---|---|---|---|---|---|---|
| | | | Novelty | Relevance | Significance | Verifiability | Avg | Pearson | Spearman |
| API-based | gpt-3.5-turbo(0-shot) | 1.90 | 1.54 | 2.69 | 1.77 | 2.08 | 2.02 | 0.87 | 0.78 |
| | gpt-3.5-turbo(5-shot)* | 1.96 | 1.31 | 2.62 | 2.08 | 2.62 | 2.15 | 0.80 | 0.78 |
| General | Llama-2-70b-chat(0-shot) | 2.04 | 1.77 | 2.23 | 1.92 | 1.92 | 1.96 | 0.89 | 0.84 |
| | Llama-2-70b-chat(5-shot) | 2.20 | 2.15 | 2.77 | 2.08 | 2.31 | 2.33 | 0.96 | 0.90 |
| | Llama-2-70b-chat(5-shot)* | 2.01 | 1.38 | 2.62 | 2.31 | 2.00 | 2.08 | 0.97 | 0.94 |
| | WizardLM-70B-V1.0(0-shot) | 1.91 | 1.38 | 2.31 | 1.54 | 2.00 | 1.81 | 0.90 | 0.75 |
| | WizardLM-70B-V1.0(5-shot) | 2.01 | 1.15 | 2.69 | 2.46 | 1.77 | 2.02 | 0.85 | 0.89 |
| Medicine | PMC-LLaMA-13B(0-shot) | 1.41 | 1.00 | 2.62 | 1.92 | 2.00 | 1.88 | 0.73 | 0.73 |
| | PMC-LLaMA-13B(5-shot)* | 1.97 | 1.85 | 2.23 | 1.92 | 1.69 | 1.92 | 0.95 | 0.94 |
| SFT | WizardLM-13BV1.2 | 1.79 | 0.85 | 2.77 | 1.23 | 2.23 | 1.77 | 0.83 | 0.85 |

Table 1: This table presents the results of human evaluation. The *Avg Coefficient* are used to assess the correlation between the average scores obtained from ChatGPT and human.

**Randomly Sampled Few-Shot Examples vs. Similarity Retrieval.** Given that randomly sampled in-context examples frequently diverge in topics or domains from the given background, this variation can potentially confuse LLMs while heightening uncertainty. In contrast, few-shot examples acquired through similarity retrieval are more likely to bolster verifiability but may reduce novelty.

**Impact of Domain Adaptation** We also conduct an analysis of the influence of fine-tuning for biomedical domain adaptation on hypothesis generation. The results obtained from MedAplaca and PMC-LLaMA indicate that domain adaptation can significantly improve word overlap performance. Evaluation derived from GPT-4 imply that domain adaptation improves the capability of LLMs in utilizing few-shot, particularly with related literature, while diminishing their zero-shot ability.

**Impact of Instruction Tuning** In addition to domain adaptation, we also examine the effect of direct fine-tuning of LLMs on specially constructed instruction training sets. The findings reveal that instruction tuning enhances the capacity of LLMs for knowledge retention, leading to improved BLEU and ROUGE scores and heightened verifiability, especially when contrasted with models without fine-tuning. However, this enhanced memorization concurrently diminishes the model's capability to generate novel and significant hypotheses.

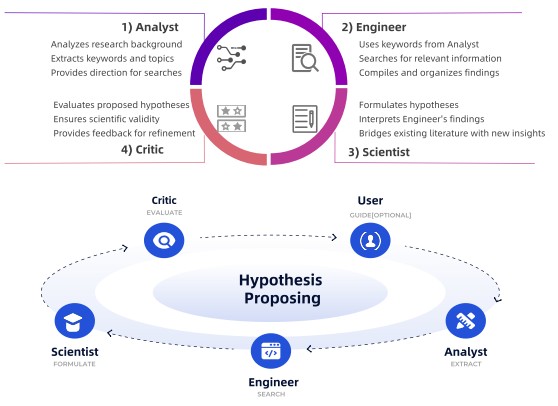

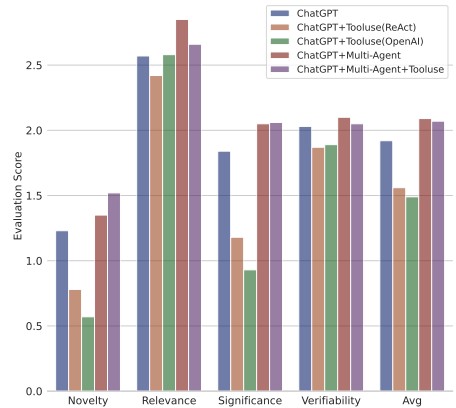

(a) Overview of Multi-agent Framework.  (b) Results of tool use and multi-agent.

Figure 6: (a) The conceptual system of multi-agent collaboration for hypothesis generation. The overall prototyping process is illustrated below, allowing users to choose optional involvement. We offer core role descriptions of multi-agents and the fully automated system above. (b) Results of individual agents and multi-agent systems, both with and without the use of tools, on the unseen test dataset. The results demonstrate the influence of multi-agent collaboration and tool use in enhancing the ability of LLMs in hypothesis generation.

## 3.3 Quantitative Analysis on Uncertainty

Drawing from the findings in zero-shot settings and the impact of external knowledge, we identify uncertainty as the key differentiator among these scenarios. The few-shot setting tends to enhance the creative capacity of LLMs, yielding content with higher generalization and diversity. In terms of external knowledge, parameterized knowledge and judiciously selected samples narrow the candidate space, thereby reducing the uncertainty of generated content but enhancing verifiability. Conversely, randomly sampled few-shot examples introduce uncertainty. There are many methods for computing uncertainty, including those based on entropy, semantics, and consistency (Kuhn et al.; Xiong et al.). However, these methods often entail a high computational cost, primarily due to the requirement for multiple sampling iterations, rendering them impractical for our experiments. Given the pioneering nature of our research in applying Large Language Models (LLMs) to knowledge discovery, and to maintain manageability within the scope of this paper, we have chosen to use SelfBLEU (Alihosseini et al., 2019) to evaluate the internal uncertainty of all generated hypotheses. The SelfBLEU scores serve as indicators of uncertainty in the hypotheses generated by LLMs, where lower scores signify greater uncertainty and diversity.

As depicted in Figure 5, we examine the relationship between uncertainty and detailed evaluation scores of various models under different settings, including zero-shot and few-shot settings (the latter comprising sampled and retrieved examples, denoted as 5-shot and 5-shot*, respectively). Our findings reveal that hypotheses generated in the zero-shot setting exhibit higher uncertainty and novelty scores, as illustrated in Figure 5a. Conversely, few-shot settings with augmented hypotheses demonstrate lower uncertainty yet higher verifiability, as indicated in Figure 5b. These observations offer valuable insights for further research into the balance between novelty and verifiability in hypothesis proposing. Motivated by these findings, we propose a LLM-based multi-agent framework to delve into uncertainty exploration involving tools and multi-agents collaboration in Section 4.

## 3.4 Human Evaluation and Case Study

In this section, we conduct human evaluation to assess the coherence of hypotheses generated by LLMs, comparing these with GPT-4 evaluation scores to guide further automated evaluations.

**Settings** We apply the four metrics from Section 3.1 (*novelty, relevance, significance, and verifiability*) for manual evaluation and comparison with GPT-4's assessments, where each metric is scored from 0 to 3. Due to cost constraints, human evaluation focuses on the highest-performing LLM based on automatic metrics and GPT-4 evaluations. The selected models and prompts are listed in Table 1. 100 examples from the unseen test set were evaluated by three experts in the biomedical field.

**Results** The results in Table 1 show a strong correlation (Pearson and Spearman coefficients above 0.7) between human and GPT-4 evaluations, reinforcing the potential of LLMs in hypothesis evaluation. For further insights, we analyze the correlation between word overlap scores and manual scores in the Appendix D. A case study illustrating various model-generated hypotheses and ChatGPT evaluations is available in Appendix E. Notably, we have collaborated with experts in medicine and biology and adopted a 5% sampling rate for expert evaluation. Given the constraints of manpower and costs, a full-scale evaluation was not feasible. However, we believe that a 5% sampling rate provides a statistically representative overview and effectively reflects the overall quality of the data. We appreciate your understanding and support in this matter.

## 4 Can agent collaboration enhance LLMs' zero-shot generalization?

In this section, we aim to improve LLMs' hypothesis generation capability by leveraging uncertainty with tool use and multi-agent collaboration. We firstly introduce the conceptual multi-agent system for hypothesis generation, drawing inspiration from scientific research.

Subsequently, we present the role design and the tools use in this context. Finally, we present preliminary validated results of the multi-agent system on the constructed dataset.

## 4.1 Multi-agent Framework

Inspired by the discovery process in Figure 2a, we introduce a comprehensive framework tailored for hypothesis formulation. This framework encapsulates a multi-agent system where each agent assumes a distinct role, mirroring the collaborative nature of scientific endeavors. Through a symbiotic and iterative process, these agents collaborate to craft hypotheses that are not only grounded in existing knowledge but also pave the way for novel insights. By emulating the essence of scientific discovery, our framework strives to produce hypotheses that are both innovative and scientifically robust. As depicted in Figure 6a, we partition the framework into five components, encompassing four automated agents and the option for human involvement within the loop.

**Role Design** In our proposed multi-agent framework, each component plays a distinct and pivotal role. The **Analyst** serves as the foundation, meticulously extracting and defining core elements from the research background. Its primary objective is to interpret the literature, distilling it into keywords or topics that subsequently guide the Engineer's search efforts. The **Engineer**, leveraging these keywords, embarks on a mission to retrieve and organize pertinent information. They meticulously plan and execute detailed searches, ensuring that the findings are compiled in a structured manner. This organized materials then lands in the domain of the **Scientist**, whose objective is to weave together the Engineer's findings with the original research background. Through careful interpretation, the Scientist crafts a hypothesis that is both grounded in existing knowledge and offers a fresh perspective. However, before this hypothesis is finalized, it undergoes scrutiny by the **Critic**. The Critic's role is paramount in ensuring the hypothesis's robustness, coherence, and novelty. They evaluate the hypothesis against the backdrop of the research background, ensuring it stands up to academic rigor. Feedback from the Critic, if necessary, loops back to refine the hypothesis or prompts the Analyst for further insights, creating a cyclical and iterative process of refinement.

**Tool Use** To explore external knowledge beyond the inherent dark knowledge within LLMs, we integrate the Engineer agent with search engines , mainly PubMed [2]. Similarly, to control the visibility of the unseen test dataset, we filter and exclude literature published after January 2023 from the search results. We carry out tool use experiments using ReAct (Yao et al., 2023b) and OpenAI function calling. ReAct is a method that extends the concept of Chain of Thought (CoT) (Wei et al., 2022), involving thinking before taking action and subsequently making observations based on feedback from the environment. In our experiments, we instruct the LLMs to initially contemplate the provided background information and then make a decision regarding whether to utilize tools. Upon receiving feedback from the tools, the LLMs are expected to identify supporting evidence in the results or potentially make further tool requests. The LLMs are responsible for concluding the hypothesis generation process and summarizing the hypotheses independently.

## 4.2 Experiment Results

Our primary focus is to investigate the impact of tool use and multi-agent collaboration on hypothesis generation. We present the experimental results in Figure 6b. Based on the results, we summarize our findings from the following two perspectives:

**Results of Tool Use** Based on our results, we observe that tool use has minimal impact on improving the hypothesis generation ability of LLMs. This observation aligns with the findings presented in Section 3.2.2 regarding the analysis of external knowledge. Notably, the ReAct-based method performs slightly worse than OpenAI function calling. It is also evident that LLMs struggle to identify useful information and exhibit weaknesses in the thought-action-observation process, even when utilizing the official interface from OpenAI.

---

[2]https://pubmed.ncbi.nlm.nih.gov/

Hypothesis generation is a demanding task where the balance between certainty and uncertainty plays a crucial role and needs further exploration.

**Results of Multi-agent Collaboration** In addition to tool use, our findings suggest that the division of labor and interaction among multi-agents can significantly enhance the model's capability to propose hypotheses by introducing uncertainty. This mirrors the dynamics of real-world scientific research, where hypotheses are formulated through iterative discussions and refutations. Additionally, it is worth noting that tool use can further enhance the performance of the multi-agent framework.

## 5 Conclusion

This study presents a comprehensive evaluation of large language models (LLMs) as biomedical hypothesis generators, focusing on their zero-shot generalization ability. We constructed a novel biomedical corpus to assess the hypothesis generation capabilities of LLMs, revealing their remarkable ability to generate novel and validated hypotheses even when tested on unseen literature. We introduced a multi-agent framework that leverages LLMs to enhance collaborative analysis in the hypothesis generation process and developed multidimensional metrics to evaluate the quality of generated hypotheses. Our work contributes to the growing research on the application of LLMs in scientific discovery, particularly in the biomedical domain. Future research could explore the integration of domain-specific knowledge, incorporation of diverse data sources, and development of user-friendly interfaces to further enhance the performance and adoption of LLMs in biomedical hypothesis generation. Our study highlights the immense potential of LLMs as catalysts for biomedical discovery, offering a promising avenue for accelerating the pace of scientific progress.

## 6 Limitations and future works

Our research reveals the potential of Large Language Models (LLMs) to generate new hypotheses, marking a significant positive finding. However, we must acknowledge certain limitations. Firstly, a thorough investigation is required into the impact of factual hallucinations inherent in LLMs on hypothesis formulation. Additionally, evaluating and analyzing knowledge discovery in LLMs demands in-depth consideration from various perspectives, necessitating further exploration.

Currently, our focus is on the hypothesis generation stage, utilizing tools such as Web search and PubMed. Introducing more specialized tools into the knowledge discovery cycle could prove invaluable. In future work, we plan to integrate bioinformatics analysis tools (Zhang et al., 2024) to enhance our research efforts.

Moving forward, we intend to incorporate knowledge graphs to aid in generating higher-quality hypotheses. Additionally, we will explore the inclusion of more dynamic tools, such as multi-tool collaborations, to further improve the performance of our LLM-based models.

To significantly enhance the hypothesis generation capabilities of our models, we aim to optimize strategies that introduce uncertainty via reinforcement learning. This approach will help us address the limitations and unlock the full potential of LLMs in the knowledge discovery process.

## Acknowledgements

This work is supported by the National Science and Technology Major Project (2023ZD0121403). We extend our gratitude to the anonymous reviewers for their insightful feedback, which has greatly contributed to the improvement of this paper.

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

# A   Process of Scientific Discovery

Scientific discovery involves key components, each crucial for advancing our understanding of the natural world: data analysis, hypothesis formulation, experiment design, execution, and observation and reflection Jain et al. (2023) as shown in Figure 2a.

1) **Data Analysis**: Foundational in the scientific process, it entails collecting and examining data to discern patterns and anomalies, extracting insights through statistical techniques and visualization. It initiates scientific inquiry, guiding further exploration. 2) **Generating Hypotheses**: Among these components, hypothesis formulation is pivotal. It entails crafting informed guesses to explain observed phenomena. Hypotheses serve as guiding frameworks, directing and focusing research by articulating specific relationships and outcomes for experimental exploration. 3) **Experiment Design**: Once a hypothesis is set, designing experiments becomes essential to rigorously test its validity. This involves defining variables, specifying control groups, and outlining methods and procedures. Well-designed experiments ensure objective hypothesis testing and yield meaningful, informative results. 4) **Experiment Execution**: Meticulous execution of designed experiments and data collection are critical. Researchers adhere precisely to experimental protocols, recording observations, measurements, and unexpected findings. Integrity in execution ensures reliable, reproducible outcomes. 5) **Accumulating Observations**: After experiments, scientists engage in observation and reflection. They analyze collected data to determine if results support or refute the initial hypothesis. If unsupported, hypotheses may be revised or new ones formulated based on findings. Observation and reflection permit iterative refinement of scientific understanding.

**Hypothesis Pioneers Pathways: Guiding Knowledge Discovery**. While all components are essential, hypothesis formulation holds a unique position. It drives the scientific endeavor, guiding research question selection, experiment design, and data analysis. Well-constructed hypotheses not only provide direction but also lay the foundation for meaningful scientific discoveries by posing rigorously testable questions. Hypothesis formulation serves as the intellectual anchor steering scientific investigation and ultimately advancing knowledge.

# B   Related Works

## B.1   Data-Driven Scientific Discovery

Data-driven knowledge discovery research within LLM is relatively limited, with the current focus primarily on dataset construction and task-driven design. In this context, (Zhong et al., 2023) proposed a dataset for investigating the transition from goals to discoveries. However, it should be noted that accurate discoveries within this dataset are not recent. (Wang et al., 2023b) introduced a method for automatically collecting and constructing publication data, along with a proposal for a hypothesis generation approach in the natural language processing (NLP) domain. However, this method requires prior human knowledge, explicit context, and is not an automated process. It's worth noting that their data was constructed from literature before 2021 from the ACL collection, implying that the information may already exist in open-source models like chatGPT and LLAMA. Furthermore, (Wang et al., 2023b) focused on integrating computational tools in the field of chemistry, primarily analyzing the capabilities of LLMs in using integrated tools but neglecting the ability for zero-shot generalization in chemistry reactions. (Boiko et al., 2023) delved more into the abilities of LLMs regarding planning and conducting experiments but did not consider proposing new hypotheses. (Yang et al., 2023b) introduced a new task for open-domain hypothesis induction and created a dataset comprising 50 articles from social science journals. Additionally, they developed a multi-module system for exploring feedback mechanisms. However, all of the above-mentioned literature lacks strict guarantees on the visibility of test data to models, thereby limiting our exploration of the zero-shot generalization capability of LLMs through learning from existing knowledge to propose new hypothesis. Unlike existing works, we have designed datasets based on publication dates, which can easily ensure a strict independence between test data and LLMs.

## B.2   LLM-driven Autonomous Agents

Large language models demonstrate exceptional capabilities in tasks such as question answering, program coding, and instruction following. However, they still confront significant challenges related to factual hallucination (Zhang et al., 2023; Rawte et al., 2023), knowledge outdated (Cao et al., 2023), and interactions with real-world. To address these challenges, recent research has explored enhancing LLMs by incorporating tools such as search engines (Nakano et al., 2022; Parisi et al., 2022), calculators (Schick et al., 2023b), code interpreter (Zhu et al., 2023b), RESTful APIs (Song et al., 2023; Patil et al., 2023) and others. The integration of LLMs with tool use, also known as LLM-driven autonomous agents (LAAs), has attracted substantial public attention. These agents are equipped with reasoning (Wei et al., 2023a; Yao et al., 2023a), planning (Shen et al., 2023; Valmeekam et al., 2023), decision-making (Yang et al., 2023a; Kang et al., 2023), and long-term memory capabilities (Zhu et al., 2023a; Hu et al., 2023), and they are constructed upon the foundation of LLMs. LAAs can autonomously plan sub-goals for complex tasks, execute actions, obtain feedback from the environment, and adjust their behaviors to adapt (Yao et al., 2023b; Xi et al., 2023a; Shinn et al., 2023). LAAs have demonstrated significant potential in addressing complex real-world tasks, including software development (Qian et al., 2023; Hong et al., 2023), drama creation (Maas et al., 2023), course design (Chen et al., 2023), chemistry experiments (Bran et al., 2023) and more. Furthermore, multi-agent collaboration plays a significant role in LAA applications, allowing agents to collaborate and interact to solve problems through various role-playing scenarios (Park et al., 2023; Fu et al., 2023; Gong et al., 2023; Li et al., 2023). To the best of our knowledge, there is still a dearth of exploration regarding the use of agents, particularly multi-agents, for scientific discovery. In this paper, our objective is to undertake a preliminary effort to enhance the hypothesis proposing capability of LLMs by harnessing tools and multiple agents, along with conducting an analysis of influencing factors.

| Category | Model | Seen | | Unseen | | | | | | |
|---|---|---|---|---|---|---|---|---|---|---|
| | | BLEU | ROUGE | BLEU | ROUGE | Novelty | Relevance | Significance | Verifiability | Avg |
| API-based | gpt-3.5-turbo(0-shot) | 13.93 | 25.32 | 15.52 | 26.48 | **1.42** | **2.63** | 1.58 | 1.97 | 1.90 |
| | gpt-3.5-turbo(5-shot) | 16.47 | 27.07 | 16.49 | 26.96 | 1.22 | 2.57 | 1.84 | 2.03 | 1.92 |
| | gpt-3.5-turbo(5-shot)* | **17.33** | **27.28** | **17.71** | **27.53** | 1.02 | 2.61 | **1.85** | **2.36** | **1.96** |
| General | Vicuna-33b-v1.3(0-shot) | 13.97 | 24.43 | 13.66 | 23.43 | 1.67 | 2.55 | 2.04 | 1.84 | 2.03 |
| | Vicuna-33b-v1.3(5-shot) | 11.23 | 22.54 | 11.49 | 22.68 | 1.60 | 2.40 | 1.67 | 1.90 | 1.89 |
| | Vicuna-33b-v1.3(5-shot)* | 12.78 | 24.11 | 13.12 | 23.66 | 1.19 | 2.71 | 2.00 | 2.17 | 2.02 |
| | Llama-2-70b-chat(0-shot) | 10.95 | 21.56 | 11.44 | 22.04 | 1.86 | 2.41 | 1.91 | 1.98 | 2.04 |
| | Llama-2-70b-chat(5-shot) | 8.17 | 21.09 | 7.63 | 20.70 | **1.95** | 2.58 | 2.06 | 2.22 | **2.20** |
| | Llama-2-70b-chat(5-shot)* | 8.40 | 21.65 | 9.66 | 22.43 | 1.43 | 2.50 | 1.94 | 2.15 | 2.01 |
| | WizardLM-13B-V1.2(0-shot) | 11.91 | 23.35 | 12.03 | 23.55 | 1.62 | 2.55 | 1.90 | 1.90 | 1.99 |
| | WizardLM-13B-V1.2(5-shot) | 14.00 | 24.30 | 13.82 | 24.38 | 1.33 | 2.54 | 1.81 | **2.23** | 1.97 |
| | WizardLM-13B-V1.2(5-shot)* | 14.96 | 25.66 | 15.26 | 25.78 | 1.06 | 2.64 | 1.73 | 2.14 | 1.89 |
| | WizardLM-70B-V1.0(0-shot) | 13.45 | 24.12 | **14.25** | **25.05** | 1.57 | 2.45 | 1.74 | 1.89 | 1.91 |
| | WizardLM-70B-V1.0(5-shot) | **14.04** | **24.59** | 13.78 | 24.28 | 1.17 | 2.61 | **2.12** | 2.14 | 2.01 |
| | WizardLM-70B-V1.0(5-shot)* | 14.46 | 24.78 | 15.26 | 25.56 | 0.97 | 2.67 | 1.85 | 1.99 | 1.87 |
| | Openchat-v3.2-super(0-shot) | 8.79 | 22.71 | 8.38 | 21.48 | 1.58 | 2.51 | 1.70 | 2.05 | 1.96 |
| | Openchat-v3.2-super(5-shot) | 12.46 | 23.60 | 12.58 | 24.21 | 1.06 | 2.64 | 2.09 | 2.20 | 2.00 |
| | Openchat-v3.2-super(5-shot)* | 12.37 | 23.93 | 12.88 | 24.78 | 1.16 | **2.76** | 2.10 | **2.23** | 2.07 |
| Medicine | MedAlpaca-13B(0-shot) | 6.10 | 22.07 | 5.82 | 20.49 | 0.55 | 1.17 | 1.17 | 1.06 | 0.99 |
| | MedAlpaca-13B(5-shot) | 0.99 | 3.84 | 1.08 | 3.84 | 0.98 | 1.32 | 1.32 | 1.49 | 1.28 |
| | MedAlpaca-13B(5-shot)* | 4.60 | 9.36 | 4.50 | 9.07 | 1.09 | 1.40 | 1.20 | 1.53 | 1.31 |
| | PMC-LLaMA-13B(0-shot) | **22.89** | **40.36** | **22.37** | **40.45** | 0.76 | 1.94 | 1.42 | 1.52 | 1.41 |
| | PMC-LLaMA-13B(5-shot) | 1.36 | 4.83 | 1.41 | 4.78 | 1.13 | 1.45 | 1.36 | 0.88 | 1.21 |
| | PMC-LLaMA-13B(5-shot)* | 6.21 | 12.39 | 6.16 | 12.13 | **1.73** | **2.17** | **1.88** | **2.09** | **1.97** |
| SFT | WizardLM-13B-V1.2 | 19.13 | 27.35 | 19.73 | 27.58 | 0.97 | 2.55 | 1.38 | 2.26 | 1.79 |

Table 2: (GPT-3.5 Datasets)Results of various LLMs: We assess instructed models using zero-shot and few-shot format prompts to generate constrained outputs. To provide a comprehensive assessment, we calculate the average scores for novelty, relevance, significance, and verifiability, denoted as Avg. Results marked with an asterisk (*) indicate that the few-shot prompts are constructed by retrieving samples from the training set that are similar to the background of inputs. To facilitate better comparison, we highlight the highest and sub-high score with both **bold** and underline formatting under each category.

| Category | Model(GPT-4) | Seen | | Unseen | | | | | | |
|---|---|---|---|---|---|---|---|---|---|---|
| | | BLEU | ROUGE | BLEU | ROUGE | Novelty | Relevance | Significance | Verifiability | Avg |
| API-based | gpt-3.5-turbo(0-shot) | 11.23 | 24.74 | 10.43 | 23.81 | **1.44** | 2.60 | **1.96** | 1.68 | 1.92 |
| | gpt-3.5-turbo(5-shot) | 13.02 | 26.01 | 12.39 | 25.18 | 0.85 | **2.79** | 1.84 | 2.26 | 1.94 |
| | gpt-3.5-turbo(5-shot)* | **13.43** | **26.22** | **12.43** | **25.60** | 0.88 | 2.68 | 1.94 | **2.40** | **1.98** |
| General | Vicuna-33b-v1.3(0-shot) | 9.75 | 22.66 | 9.61 | 22.45 | 1.80 | 2.67 | 1.77 | 1.75 | 2.00 |
| | Vicuna-33b-v1.3(5-shot) | 9.43 | 22.66 | 9.71 | 22.57 | 1.05 | 2.43 | 2.41 | 1.70 | 1.90 |
| | Vicuna-33b-v1.3(5-shot)* | 10.49 | 23.91 | 10.36 | 24.04 | 1.48 | 2.48 | 2.27 | 2.00 | 2.06 |
| | Llama-2-70b-chat(0-shot) | 9.38 | 21.95 | 8.23 | 20.8 | 1.82 | 2.67 | 2.09 | 1.88 | 2.11 |
| | Llama-2-70b-chat(5-shot) | 5.73 | 20.28 | 5.68 | 19.40 | **2.13** | 2.63 | 2.37 | 2.13 | **2.31** |
| | Llama-2-70b-chat(5-shot)* | 7.50 | 21.32 | 6.62 | 20.88 | 2.05 | 2.67 | 2.15 | 2.04 | 2.23 |
| | WizardLM-13B-V1.2(0-shot) | 9.22 | 22.56 | 8.31 | 21.47 | 1.63 | 2.72 | 1.91 | 1.88 | 2.03 |
| | WizardLM-13B-V1.2(5-shot) | 11.24 | 24.62 | 10.82 | 23.91 | 0.95 | 2.40 | 2.17 | 2.29 | 1.95 |
| | WizardLM-13B-V1.2(5-shot)* | **12.23** | **25.53** | **11.55** | **24.93** | 0.62 | 2.68 | 2.30 | 1.96 | 1.89 |
| | WizardLM-70B-V1.0(0-shot) | 10.71 | 24.04 | 10.08 | 22.54 | 1.92 | 2.36 | 2.00 | 1.87 | 2.04 |
| | WizardLM-70B-V1.0(5-shot) | 11.23 | 24.46 | 10.48 | 23.55 | 1.10 | 2.91 | 2.06 | 2.00 | 2.01 |
| | WizardLM-70B-V1.0(5-shot)* | 11.78 | 24.72 | 11.32 | 24.71 | 1.05 | 2.64 | 2.09 | 2.20 | 1.99 |
| | Openchat-v3.2-super(0-shot) | 7.24 | 22.11 | 6.96 | 21.37 | 1.68 | 2.88 | 1.88 | 2.38 | 2.20 |
| | Openchat-v3.2-super(5-shot) | 9.50 | 23.34 | 9.61 | 22.84 | 1.44 | **2.92** | 1.90 | 2.13 | 2.10 |
| | Openchat-v3.2-super(5-shot)* | 9.43 | 23.52 | 9.46 | 23.29 | 0.76 | 2.83 | 2.30 | 2.36 | 2.06 |
| Medicine | MedAlpaca-13B(0-shot) | 1.52 | 9.56 | 1.49 | 9.26 | 0.36 | 1.32 | 0.36 | 1.16 | 0.80 |
| | MedAlpaca-13B(5-shot) | 4.51 | 9.62 | 4.00 | 9.86 | 1.36 | 1.32 | 0.66 | 1.37 | 1.18 |
| | MedAlpaca-13B(5-shot)* | 2.98 | 10.33 | 3.04 | 10.66 | **1.75** | 1.32 | 1.00 | 1.84 | 1.48 |
| | PMC-LLaMA-13B(0-shot) | **8.35** | **18.68** | **8.32** | **19.33** | 0.58 | 2.28 | 1.08 | 2.40 | 1.59 |
| | PMC-LLaMA-13B(5-shot) | 1.01 | 4.47 | 1.01 | 4.55 | 0.64 | 2.20 | 1.33 | 2.38 | 1.64 |
| | PMC-LLaMA-13B(5-shot)* | 6.96 | 15.59 | 1.20 | 5.27 | 0.68 | **2.32** | **1.52** | **2.64** | **1.79** |
| SFT | WizardLM-13B-V1.2 | 18.11 | 25.65 | 18.44 | 26.98 | 1.01 | 2.46 | 1.40 | 2.20 | 1.77 |

Table 3: (GPT-4 datasets) Results of various LLMs: We assess instructed models using zero-shot and few-shot format prompts to generate constrained outputs. To provide a comprehensive assessment, we calculate the average scores for novelty, relevance, significance, and verifiability, denoted as Avg. Results marked with an asterisk (*) indicate that the few-shot prompts are constructed by retrieving samples from the training set that are similar to the background of inputs. To facilitate better comparison, we highlight the highest and sub-high score with both **bold** and underline formatting under each category.

## C  Implementation Details

In this section, we delve into further implementation details of our experiments, including information about the constructed dataset and open-source models.

### C.1  Details of Dataset

We present the publication dates and topic distributions of the various datasets for comparison, as illustrated in Figure 7, where we utilize `Nomic Atlas` [3] to visualize the topic distribution of abstracts in both the training and test datasets.

As depicted in Figure 8a, we embed the background and hypothesis sentences from both the seen and unseen test sets using OpenAI embeddings , followed by dimensionality reduction through t-SNE (Van der Maaten & Hinton, 2008). The results indicate minimal differences between the two test sets, despite their derivation from distinct time periods.

### C.2  Details of Models

We present the meta-information of the open-source models used in our experiments, as shown in Table 4. We have gathered data regarding their pre-training, supervised learning corpus, and release dates to ensure the non-visibility of the unseen test data. The models were selected based on their performance ranking on the Alpaca-Eval leaderboard prior to September 2023.

---

[3] https://github.com/nomic-ai/nomic

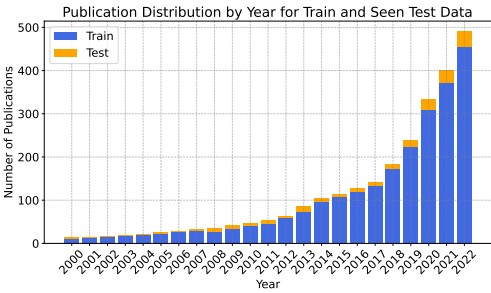 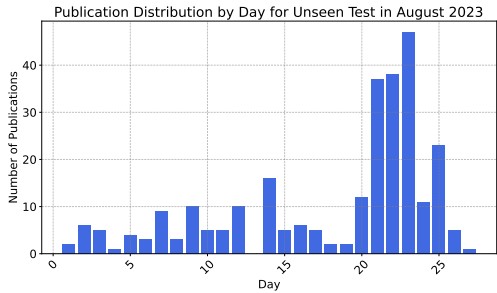

Figure 7: Distribution of the background and hypothesis pairs (BHP) dataset: we present the publication distribution by year for the training and seen test datasets, indicating a steady increase year by year until January 2023. In the center panel, we depict the publication distribution by month for the unseen test dataset, which was sampled from August 2023 and emphasizes the latter part of the month.

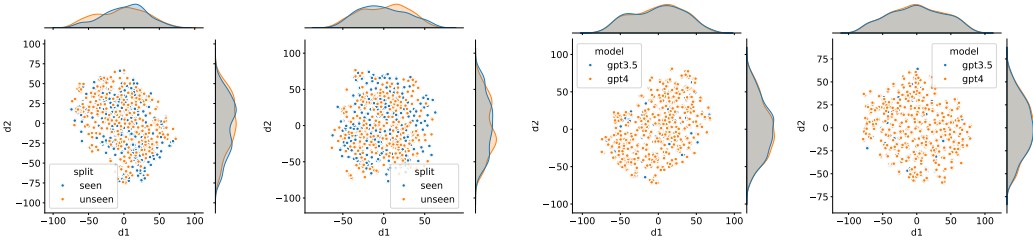

(a) Distribution of seen and unseen datasets.  (b) Distribution of ChatGPT and GPT-4 data.

Figure 8: Distribution of background (left in (a) and (b)) and hypothesis (right in (a) and (b)) texts. (a): A slight difference is observed between the distributions of seen and unseen datasets, attributable to the differences in publication dates. (b): The distribution of datasets constructed by ChatGPT and GPT-4 shows minimal variance.

## D   Additional Results

### D.1   Automatic Evaluations

We present more detailed results of diverse LLMs and prompting scenarios on datasets specifically constructed for ChatGPT and GPT-4 in Table 2 and Table 3, respectively. Additionally, we provide the outcomes of experiments involving multi-agent interactions and tool use in Table 6. In addition to Figure 3, Figure 9 presents the ROUGE scores for both seen and unseen datasets.

### D.2   Human Evaluations

We have included additional results from human evaluations in Table 5, primarily focusing on correlation scores between word overlap metrics and manual evaluations. Note that we continue to use the same samples used in human evaluation to compute BLEU and ROUGE-L for a fair comparison. We calculate the Pearson and Spearman coefficients between each automatic metric and the average human score. These results reveal that word overlap metrics, such as BLEU and ROUGE-L, exhibit notably lower correlation with manual scores. While BLEU and ROUGE-L may have a high correlation with relevance metrics, they are weak in providing a comprehensive evaluation of the generations. Conversely, evaluations conducted by ChatGPT demonstrate higher correlation with human evaluations, as illustrated in Table 1. However, there is still a significant need to explore advanced metrics, particularly automated ones, in the context of scientific discovery.

| Category | Model | Base Model | SFT Data (Y/M) | Released |
|---|---|---|---|---|
| API-based | gpt-3.5-turbo (0-shot) | GPT-3 | Unknown | 2022/12 |
| | gpt-3.5-turbo (5-shot) | GPT-3 | Unknown | 2022/12 |
| | gpt-4* | GPT-4 | Unknown | 2023/06 |
| General | Vicuna-33b-v1.3 | Llama-1 | ShareGPT (Unknown) | 2023/06 |
| | Llama-2-7b-chat | Llama-2 | Unknown | 2023/07 |
| | Llama-2-13b-chat | Llama 2 | Unknown | 2023/07 |
| | Llama-2-70b-chat | Llama 2 | Unknown | 2023/07 |
| | WizardLM-13B-V1.2 | Llama-2 | Alpaca and ShareGPT (2023/06) | 2023/07 |
| | WizardLM-70B-V1.0 | Llama-2 | Alpaca and ShareGPT (2023/06) | 2023/08 |
| | openchat-v3.2-super | Llama-2 | Sharegpt4 Dataset (2023/06) | 2023/09 |
| Medicine | MedAlpaca-13B | Llama-1* | Mixture (2023/03) | 2023/03 |
| | ChatDoctor* | Llama-1* | Mixture (2023/04) | 2023/04 |
| | PMC-LLaMA-13B | Llama-2* | Mixture (2023/04) | 2023/08* |

Table 4: To further ensure the non-visibility of the test data, we provide an overview of the related literature corpus within the training set of various LLMs, accompanied by their respective publication dates. The data marked with (*) is the data generated by people talking to ChatGPT. Our date marking is consistent with ChatGPT.

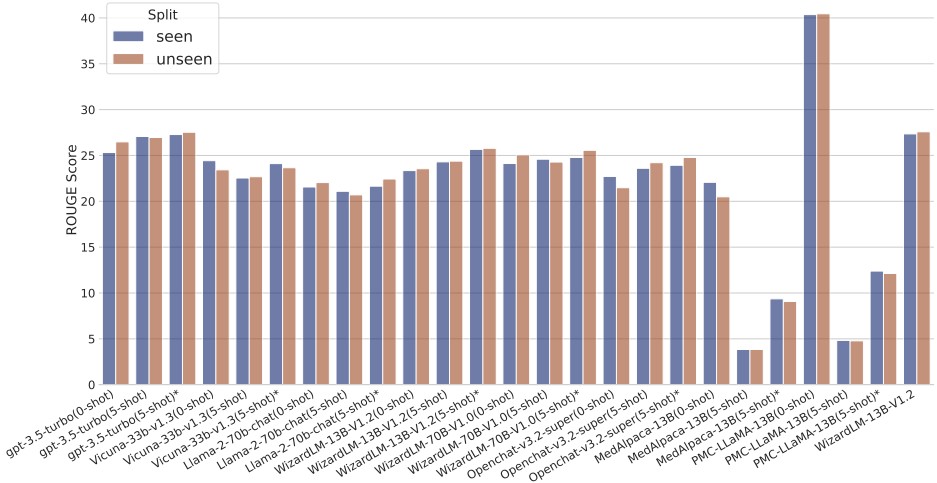

Figure 9: This figure displays the ROUGE Scores on both seen and unseen datasets.

# E   Case Study

In this section, we present several generated hypotheses from various models and provide examples of the evaluation process, step by step, using ChatGPT.

## E.1   Generated Hypothesis

We compare the generated hypotheses of different LLMs selected in human evaluation. The selected medicine literature was published in August 2023 (Chang et al., 2023), which proposed the power law index as an early marker of myocardial infarction. As shown in Table 7, some responses like `gpt-3.5-turbo (0-shot)` propose this hypothesis from zero, based only on the background. This indicates that LLMs have great potential in proposing highly novel and verifiable hypotheses and could play a significant role in scientific discovery. The case study focuses on MI predictive biomarkers and two-stage power-law rheology. GPT-3.5 identifies a "fundamental biomechanical principle," while `Llama-2-70b-chat` discusses "changes in the levels of collagen and proteoglycans in the extracellular matrix," and `WizardLM-70B-V1.0` addresses "the progression of myocardial

---

[4]https://openai.com/blog/function-calling-and-other-api-updates

| Category | Model | Word Overlap | | ChatGPT | Human |
|---|---|---|---|---|---|
| | | BLEU ($r/\rho$) | ROUGE-L ($r/\rho$) | Avg ($r/\rho$) | Avg ($r/\rho$) |
| API-based | gpt-3.5-turbo(0-shot) | 16.59(0.03/0.01) | 29.87(-0.04/-0.05) | 1.90(0.87/0.78) | 2.02(1.00/1.00) |
| | gpt-3.5-turbo(5-shot)* | 14.99(-0.09/0.12) | 27.51(-0.33/-0.35) | 1.96(0.80/0.78) | 2.15(1.00/1.00) |
| General | Llama-2-70b-chat(0-shot) | 9.64(-0.21/-0.20) | 22.17(-0.31/-0.28) | 2.04(0.89/0.84) | 1.96(1.00/1.00) |
| | Llama-2-70b-chat(5-shot) | 9.42(-0.58/-0.65) | 20.59(-0.47/-0.42) | 2.20(0.96/0.90) | 2.33(1.00/1.00) |
| | Llama-2-70b-chat(5-shot)* | 9.60(-0.16/-0.10) | 19.99(-0.15/-0.17) | 2.01(0.97/0.94) | 2.08(1.00/1.00) |
| | WizardLM-70B-V1.0(0-shot) | 11.42(0.21/0.36) | 24.11(0.29/0.49) | 1.91(0.90/0.75) | 1.81(1.00/1.00) |
| | WizardLM-70B-V1.0(5-shot) | 9.86(-0.28/-0.37) | 23.52(-0.17/-0.24) | 2.01(0.85/0.89) | 2.02(1.00/1.00) |
| Medicine | PMC-LLaMA-13B(0-shot) | 8.19(0.32/0.39) | 21.85(0.18/0.27) | 1.41(0.73/0.73) | 1.88(1.00/1.00) |
| | PMC-LLaMA-13B(5-shot)* | 5.52(0.06/-0.01) | 13.64(0.26/0.23) | 1.97(0.95/0.94) | 1.92(1.00/1.00) |
| SFT | WizardLM-13B-V1.2 | 21.48(-0.00/0.00) | 27.83(0.17/0.27) | 1.79(0.83/0.85) | 1.77(1.00/1.00) |

Table 5: The table illustrates the correlations between automatic metrics and human evaluations. We annotate the Pearson and Spearman scores after each correlation score, denoting them as $r$ and $\rho$.

| Model | Influence Factor | | Automatic | | GPT-4 Evaluation | | | | |
|---|---|---|---|---|---|---|---|---|---|
| | Multi-agent | Tool use | BLUE | ROUGE | Novelty | Relevance | Significance | Verifiability | Avg |
| 1 | - | - | 15.52 | **26.48** | 1.23 | 2.57 | 1.84 | 2.03 | 1.92 |
| 2[a] | - | ✓ | 14.94 | 24.16 | 0.78 | 2.42 | 1.18 | 1.87 | 1.56 |
| 2[b] | - | ✓ | **15.87** | 24.94 | 0.57 | 2.58 | 0.93 | 1.89 | 1.49 |
| 3 | ✓ | - | 11.71 | 22.11 | 1.35 | **2.85** | 2.05 | **2.10** | **2.09** |
| 4 | ✓ | ✓ | 11.18 | 22.04 | **1.52** | 2.66 | **2.06** | 2.05 | 2.07 |

Table 6: Results of individual agents and multi-agent systems, both with and without the use of tools, on the unseen test dataset. The results demonstrate that both multi-agent systems and the utilization of tools enhance the ability of LLMs in hypothesis generation. Among the various types of models, both 2[a] and 2[b] are evaluated with tool use. The difference between them lies in their implementations: ReAct (Yao et al., 2023b) and OpenAI function calling [4], respectively.

damage and remodeling." These models not only propose hypotheses that align with the relevant background but also offer novel insights, demonstrating a capacity for originality beyond mere information extraction. In contrast, PMC-LLaMA-13B and WizardLM-13B-V1.2 provided only straightforward interpretations of the results, without introducing new content. Furthermore, we showcase the proposed hypotheses in a multi-agent collaboration involving three roles: Analyst, Scientist, and Critic, working together to generate hypotheses across 6 rounds, as shown in Table 8.

---

INSTRUCTED LLMS CASE

**Background:**
(1) In the early stage of MI, $\alpha$short and $\alpha$long can track and distinguish MI samples from healthy control samples with an AUC > 0.7.
(2) In the mid-stage and late-stage of MI, E2, E3, and Etotal exhibit excellent discriminating performance with an AUC > 0.6.
(3) All cardiac tissues display a universal two-stage power-law rheology at different time scales, independent of their disease status.

**Golden Hypothesis:**
(1) The sequential order of MI-related changes in cardiac tissues starts from the cytoplasmic level and progresses to the cellular and tissue levels.
(2) Pathological changes at subcellular, cellular, and tissue levels are accompanied by mechanical alterations.
(3) The proposed mechanical markers have diagnostic potential for assessing the progression of MI and studying other cardiovascular diseases.

---

**gpt-3.5-turbo (0-shot):**
(1) The combination of $\alpha$short and $\alpha$long can be used as a reliable biomarker for early detection and differentiation of myocardial infarction (MI) from healthy controls.
(2) E2, E3, and Etotal can serve as effective biomarkers for distinguishing between mid-stage and late-stage MI patients with high accuracy.
(3) The universal two-stage power-law rheology observed in cardiac tissues may play a crucial role in maintaining tissue integrity and function, regardless of the presence or absence of disease.

---

**gpt-3.5-turbo (5-shot):**
(1) The levels of $\alpha$short and $\alpha$long may serve as potential biomarkers for early detection and diagnosis of myocardial infarction (MI).
(2) E2, E3, and Etotal may have utility as biomarkers for assessing the progression and severity of MI in the mid and late stages.
(3) The universal two-stage power-law rheology observed in cardiac tissues suggests a fundamental biomechanical principle underlying tissue behavior, regardless of disease status.

---

**Llama-2-70b-chat (0-shot):**
(1) E2, E3, and Etotal can also track and distinguish MI samples from healthy control samples in the early stage of MI, with an AUC > 0.7, similar to $\alpha$short and $\alpha$long.
(2) The universal two-stage power-law rheology of cardiac tissues is related to the alteration of the extracellular matrix and the degradation of collagen in the early stage of MI, which can be detected by E2, E3, and Etotal.
(3) The discriminating performance of E2, E3, and Etotal in the mid-stage and late-stage of MI is affected by the changes in the levels of collagen and proteoglycans in the extracellular matrix, which can be evaluated by measuring the mechanical properties of cardiac tissues using a uniaxial tensiometer.

---

**Llama-2-70b-chat (5-shot):**
(1) The discriminating performance of $\alpha$short and $\alpha$long in the early stage of MI is attributed to their ability to capture the subtle changes in cardiac tissue mechanics caused by the onset of fibrosis.
(2) E2, E3, and Etotal exhibit excellent discriminating performance in the mid-stage and late-stage of MI due to their sensitivity to changes in collagen content and cross-linking, which are hallmarks of fibrosis progression.

(3) The universal two-stage power-law rheology observed in all cardiac tissues, independent of disease status, suggests a common mechanism of mechanical regulation that can be exploited for early disease detection and monitoring.

**Llama-2-70b-chat (5-shot)*:**
(1) The $\alpha$short and $\alpha$long parameters can detect changes in myocardial stiffness in the early stages of myocardial infarction (MI).
(2) E2, E3, and Etotal can differentiate between healthy controls and MI samples in the mid and late stages.
(3) The universal two-stage power-law rheology in cardiac tissues may be useful in developing novel therapeutic approaches for MI.

**WizardLM-70B-V1.0 (0-shot):**
(1) $\alpha$short and $\alpha$long can track and distinguish MI samples from healthy control samples in all stages of MI with an AUC > 0.7.
(2) E2, E3, and Etotal exhibit excellent discriminating performance in the early stage of MI with an AUC > 0.6.
(3) The two-stage power-law rheology observed in all cardiac tissues is related to the progression of MI.

**WizardLM-70B-V1.0 (5-shot):**
(1) The discriminating performance of $\alpha$short and $\alpha$long in the early stage of MI may be attributed to their sensitivity to early molecular changes in the myocardium.
(2) The discriminating performance of E2, E3, and Etotal in the mid-stage and late-stage of MI may be due to their ability to reflect the progression of myocardial damage and remodeling.
(3) The universal two-stage power-law rheology observed in all cardiac tissues may be a fundamental property of cardiac tissue biomechanics, potentially useful for developing diagnostic and therapeutic strategies for MI.

**PMC-LLaMA-13B (0-shot):**
(1) $\alpha$short and $\alpha$long are effective biomarkers for distinguishing MI samples from healthy control samples both in the early stage and the late stage of MI.
(2) E2 and E3 (Etotal) are likely to serve as effective biomarkers for discriminating mid-stage or late-stage MI samples from healthy control samples.
(3) Cardiac tissues display a universal two-stage power-law rheology that is independent of their disease status.

**PMC-LLaMA-13B (5-shot):**
(1) Further comprehensive validation of the proposed framework is needed to elucidate the pathological changes in MI patients.
(2) Consideration of multiple markers is important.
(3) The influence of various external factors should be incorporated for accurate interpretation of the mechanical markers.

**WizardLM-13B-V1.2 (SFT):**
(1) The combination of $\alpha$short and $\alpha$long features can provide a more robust identification of MI samples.
(2) E2, E3, and Etotal features can serve as independent predictors of MI.
(3) The universal two-stage power-law rheology observed in all cardiac tissues may have diagnostic implications.

Table 7: This table illustrates the hypothesis proposals from different LLMs.

**MULTI-AGENT CASE**
**Background:**

(1) The past 25 years have seen significant development in the field of digital electrophysiology, including advancements in telemonitoring of cardiac implantable electronic devices (CIEDs) and the use of wearable devices for cardiac rhythm monitoring.
(2) Telemonitoring of CIEDs has been shown to reduce time to detection of clinically actionable events and improve patient outcomes.
(3)Wearable devices, such as smartwatches, rings, and fitness trackers, have become more accurate and reliable in monitoring cardiac rhythm using photoplethysmography.

**Golden Hypothesis:**
(1) The implementation of telemonitoring and the use of wearable devices for cardiac rhythm monitoring will continue to increase in the future.
(2) Artificial intelligence can be utilized to better identify actionable alerts and improve patient outcomes in telemonitoring of CIEDs.
(3) Smartwatches may have the potential to replace CIEDs in certain cases.

**Generated Hypothesis:**
(1) The advancements in telemonitoring of cardiac implantable electronic devices (CIEDs) and the use of wearable devices for cardiac rhythm monitoring have led to improved patient outcomes, such as reduced hospitalizations, improved quality of life, and decreased mortality rates.
(2) These advancements achieve these outcomes by enabling early detection of abnormal rhythms, facilitating timely interventions, and providing valuable data for personalized treatment plans.
(3) This hypothesis is supported by studies that have demonstrated the effectiveness of telemonitoring in reducing time to detection of clinically actionable events and improving patient outcomes.The refined hypothesis captures the essence of the research background and the collaborative insights, providing a logical and scientifically grounded statement. It can now be further evaluated and tested in future research.

**Dialogue process of multi-agent collaboration.**
*(Round 1) Analyst:* Based on the provided research background, I have identified the following keywords and topics:

Keywords:
- Digital electrophysiology
- Telemonitoring
- Cardiac implantable electronic devices (CIEDs)
- Wearable devices
- Cardiac rhythm monitoring
- Photoplethysmography

Topics:
1. Advancements in digital electrophysiology and telemonitoring of CIEDs
2. Benefits of telemonitoring in improving patient outcomes
3. Accuracy and reliability of wearable devices for cardiac rhythm monitoring using photoplethysmography

Preliminary insights:
- The field of digital electrophysiology has seen significant development in the past 25 years, particularly in the areas of telemonitoring of CIEDs and the use of wearable devices for cardiac rhythm monitoring.
- Telemonitoring of CIEDs has been shown to have positive impacts, such as reducing the time to detect clinically actionable events and improving patient outcomes.
- Wearable devices, including smartwatches, rings, and fitness trackers, have become more accurate and reliable in monitoring cardiac rhythm using photoplethysmography.

Based on these insights, it can be hypothesized that the advancements in telemonitoring and wearable devices have led to improved cardiac rhythm monitoring and patient outcomes. However, further analysis and collaboration with the Scientist and Critic are needed to refine and validate this hypothesis.

**(Round 2) Scientist:** Based on the insights provided by the Analyst and the research background, I propose the following hypothesis:

Hypothesis: The advancements in telemonitoring of cardiac implantable electronic devices (CIEDs) and the use of wearable devices for cardiac rhythm monitoring have led to improved patient outcomes and more accurate detection of clinically actionable events.

This hypothesis suggests that the development and implementation of telemonitoring technologies for CIEDs, along with the increasing accuracy and reliability of wearable devices using photoplethysmography, have positively impacted patient outcomes by reducing the time to detect clinically actionable events. By continuously monitoring cardiac rhythm, these advancements have the potential to provide timely interventions and improve overall patient care.

The hypothesis is grounded in the existing literature, which highlights the benefits of telemonitoring in improving patient outcomes and the increasing accuracy of wearable devices for cardiac rhythm monitoring. However, further collaboration with the Critic is necessary to evaluate the novelty, coherence, and scientific validity of the hypothesis.

**(Round 3) Critic:** The hypothesis proposed by the Scientist is well-aligned with the research background and the insights provided by the Analyst. It effectively captures the essence of the advancements in telemonitoring of CIEDs and the use of wearable devices for cardiac rhythm monitoring. The hypothesis also highlights the potential positive impact on patient outcomes and the detection of clinically actionable events.

However, there are a few areas that could be further refined or expanded upon in the hypothesis:

1. **Specific outcomes**: The hypothesis mentions ""improved patient outcomes,"" but it would be beneficial to specify the specific outcomes that are expected to improve. For example, are we referring to reduced hospitalizations, improved quality of life, or decreased mortality rates?

2. **Mechanisms**: The hypothesis briefly mentions the potential for timely interventions and improved patient care, but it would be helpful to elaborate on the specific mechanisms through which the advancements in telemonitoring and wearable devices are expected to achieve these outcomes. Are there specific interventions or treatments that can be administered based on the data collected from these devices?

3. **Scientific validity**: While the hypothesis is grounded in the existing literature, it would be valuable to provide some references or studies that support the claims made in the hypothesis. This would enhance the scientific validity of the hypothesis and demonstrate that it is not just speculative but based on empirical evidence.

To refine the hypothesis, I would recommend the Scientist to consider incorporating these suggestions and providing more specific details and references to support the claims made. This will strengthen the coherence and scientific validity of the hypothesis.

Analyst, based on the feedback provided, please re-evaluate the research background and provide any additional insights or interpretations that could further refine the hypothesis.

**(Round 4) Analyst:** Based on the feedback from the Critic, I have re-evaluated the research background and identified additional insights:

1. **Specific outcomes**: The research background mentions that telemonitoring of CIEDs has been shown to reduce time to detection of clinically actionable events and improve patient outcomes. These outcomes could include early detection of arrhythmias, reduced hospitalizations due to cardiac events, improved quality of life, and decreased mortality rates. It would be beneficial to specify one or more of these specific outcomes in the hypothesis.

2. **Mechanisms**: The advancements in telemonitoring and wearable devices can potentially lead to improved patient outcomes through several mechanisms. For example, continuous monitoring of cardiac rhythm can enable early detection of abnormal rhythms, allowing for timely interventions such as medication adjustments or electrical cardioversion. Additionally, the data collected from these devices can provide valuable insights for personalized treatment plans and remote patient management. Including these specific mechanisms in the hypothesis would enhance its coherence.

3. **Scientific validity**: To strengthen the scientific validity of the hypothesis, it would be valuable to reference specific studies or research that support the claims made. For example, citing studies that have demonstrated the effectiveness of telemonitoring in reducing time to detection of clinically actionable events or improving patient outcomes would provide empirical evidence to support the hypothesis.

Incorporating these additional insights and suggestions, the refined hypothesis could be:

Hypothesis: The advancements in telemonitoring of cardiac implantable electronic devices (CIEDs) and the use of wearable devices for cardiac rhythm monitoring have led to improved patient outcomes, such as reduced hospitalizations, improved quality of life, and decreased mortality rates. These advancements achieve these outcomes by enabling early detection of abnormal rhythms, facilitating timely interventions, and providing valuable data for personalized treatment plans. This hypothesis is supported by studies that have demonstrated the effectiveness of telemonitoring in reducing time to detection of clinically actionable events and improving patient outcomes.

This refined hypothesis takes into account the specific outcomes, mechanisms, and scientific validity, as suggested by the Critic. It provides a more comprehensive and well-supported statement that captures the essence of the research background and the collaborative insights.

*(Round 5) Scientist:* Based on the additional insights and suggestions provided by the Analyst and the Critic, I propose the following refined hypothesis:

Hypothesis: The advancements in telemonitoring of cardiac implantable electronic devices (CIEDs) and the use of wearable devices for cardiac rhythm monitoring have led to improved patient outcomes, such as reduced hospitalizations, improved quality of life, and decreased mortality rates. These advancements achieve these outcomes by enabling early detection of abnormal rhythms, facilitating timely interventions, and providing valuable data for personalized treatment plans. This hypothesis is supported by studies that have demonstrated the effectiveness of telemonitoring in reducing time to detection of clinically actionable events and improving patient outcomes.

This refined hypothesis incorporates the specific outcomes mentioned in the research background, such as reduced hospitalizations, improved quality of life, and decreased mortality rates. It also expands on the mechanisms through which these advancements are expected to achieve these outcomes, including early detection of abnormal rhythms, timely interventions, and personalized treatment plans. Additionally, the hypothesis acknowledges the importance of scientific validity by referencing studies that support the claims made.

The refined hypothesis is both innovative and scientifically grounded, as it builds upon the existing literature and incorporates the collaborative insights provided by the Analyst and the Critic. It provides a logical and well-supported statement that can be further evaluated and tested.

*(Round 6) Critic:* The refined hypothesis proposed by the Scientist is well-structured and addresses the feedback provided by the Critic. It incorporates specific outcomes, mechanisms, and scientific validity, making it a comprehensive and scientifically robust statement.

Final Answer:
(1) The advancements in telemonitoring of cardiac implantable electronic devices (CIEDs) and the use of wearable devices for cardiac rhythm monitoring have led to improved patient outcomes, such as reduced hospitalizations, improved quality of life, and decreased mortality rates.
(2) These advancements achieve these outcomes by enabling early detection of abnormal rhythms, facilitating timely interventions, and providing valuable data for personalized treatment plans.
(3) This hypothesis is supported by studies that have demonstrated the effectiveness of telemonitoring in reducing time to detection of clinically actionable events and improving patient outcomes.

The refined hypothesis captures the essence of the research background and the collaborative insights, providing a logical and scientifically grounded statement. It can now be further evaluated and tested in future research.

Table 8: This table illustrates a case from multi-agent collaboration on hypothesis proposing.

E.2    Evaluation Examples

We provide evaluation results conducted by ChatGPT in Table 9. ChatGPT offers a step-by-step explanation for each metric, ultimately providing independent scores in integer format.

---

**CASE OF CHATGPT AS ANNOTATOR**
**Background:**
(1) The past 25 years have seen significant development in the field of digital electrophysiology, including advancements in telemonitoring of cardiac implantable electronic devices (CIEDs) and the use of wearable devices for cardiac rhythm monitoring.
(2) Telemonitoring of CIEDs has been shown to reduce time to detection of clinically actionable events and improve patient outcomes.
(3) Wearable devices, such as smartwatches, rings, and fitness trackers, have become more accurate and reliable in monitoring cardiac rhythm using photoplethysmography.

**Golden Hypothesis:**
(1) The implementation of telemonitoring and the use of wearable devices for cardiac rhythm monitoring will continue to increase in the future.
(2) Artificial intelligence can be utilized to better identify actionable alerts and improve patient outcomes in telemonitoring of CIEDs.
(3) Smartwatches may have the potential to replace CIEDs in certain cases.

**Generated Hypothesis:**
(1) The advancements in telemonitoring of cardiac implantable electronic devices (CIEDs) and the use of wearable devices for cardiac rhythm monitoring have led to improved patient outcomes, such as reduced hospitalizations, improved quality of life, and decreased mortality rates.

(2) These advancements achieve these outcomes by enabling early detection of abnormal rhythms, facilitating timely interventions, and providing valuable data for personalized treatment plans.

(3) This hypothesis is supported by studies that have demonstrated the effectiveness of telemonitoring in reducing time to detection of clinically actionable events and improving patient outcomes.The refined hypothesis captures the essence of the research background and the collaborative insights, providing a logical and scientifically grounded statement. It can now be further evaluated and tested in future research.

**Evaluation Results**

**Novelty:**

Explanation: The generated scientific hypothesis builds upon the background information by stating that the advancements in telemonitoring of CIEDs and the use of wearable devices for cardiac rhythm monitoring have led to improved patient outcomes. It further suggests that these advancements achieve these outcomes by enabling early detection of abnormal rhythms, facilitating timely interventions, and providing valuable data for personalized treatment plans. This hypothesis goes beyond the background by proposing specific mechanisms through which the advancements in telemonitoring and wearable devices contribute to improved patient outcomes. It also mentions the supporting evidence from studies that have demonstrated the effectiveness of telemonitoring in reducing time to detection of clinically actionable events and improving patient outcomes. Overall, the hypothesis provides new insights and expands upon the existing knowledge in the field.

Score: 2

**Relevance:**

Explanation: The generated scientific hypothesis is strongly related to the given background. The background provides information about the advancements in telemonitoring of cardiac implantable electronic devices (CIEDs) and the use of wearable devices for cardiac rhythm monitoring. The hypothesis builds upon this information by stating that these advancements have led to improved patient outcomes, such as reduced hospitalizations, improved quality of life, and decreased mortality rates. It further explains that these outcomes are achieved by enabling early detection of abnormal rhythms, facilitating timely interventions, and providing valuable data for personalized treatment plans. The hypothesis is supported by the background information, which mentions that telemonitoring of CIEDs has been shown to reduce time to detection of clinically actionable events and improve patient outcomes. Overall, the hypothesis is directly related to the background and provides a logical and scientifically grounded statement.

Score: 3

**Significance:**

Explanation: The generated scientific hypothesis builds upon the background information provided and presents a logical and scientifically grounded statement. It suggests that the advancements in telemonitoring of CIEDs and the use of wearable devices for cardiac rhythm monitoring have led to improved patient outcomes, such as reduced hospitalizations, improved quality of life, and decreased mortality rates. The hypothesis further suggests that these advancements achieve these outcomes by enabling early detection of abnormal rhythms, facilitating timely interventions, and providing valuable data for personalized treatment plans. This hypothesis is supported by the background information that mentions the effectiveness of telemonitoring in reducing time to detection of clinically actionable events and improving patient outcomes. Overall, the hypothesis provides significant insights beyond the background and presents a testable statement for future research.

Score: 3

**Verifiability:**

Explanation: The generated scientific hypothesis is highly verifiable based on the given background. The background provides evidence of significant advancements in telemonitoring of cardiac implantable electronic devices (CIEDs) and the use of wearable devices for cardiac rhythm monitoring. It also states that telemonitoring of CIEDs has been shown to reduce time to detection of clinically actionable events and improve patient outcomes. Additionally, it mentions that wearable devices have become more accurate and reliable in monitoring cardiac rhythm.

The hypothesis builds upon this background by suggesting that these advancements have led to improved patient outcomes, such as reduced hospitalizations, improved quality of life, and decreased mortality rates. It further proposes that these outcomes are achieved by enabling early detection of abnormal rhythms, facilitating timely interventions, and providing valuable data for personalized treatment plans.

The hypothesis is supported by studies that have demonstrated the effectiveness of telemonitoring in reducing time to detection of clinically actionable events and improving patient outcomes. Therefore, the hypothesis is highly verifiable and has a strong likelihood of being confirmed in future research.
Score: 3

Table 9: This table illustrates a case for hypothesis evaluation using ChatGPT.

## F    Prompts Design and Finetuning

In this section, we offer prompts for prompt LLMs for hypothesis generation and for ChatGPT in the evaluation process.

### F.1    Zero- and Few-shot Prompts

We provide prompts for hypothesis generation under both zero-shot and few-shot settings, which are detailed in Table 10 and Table 11 respectively. Specially, the latter includes two methods to obtain in-context examples: random sampling and similarity retrieval.

---

**ZERO-SHOT INSTRUCTION FOR HYPOTHESIS GENERATION.**
You are a researcher. You can give novel hypothesis based on your exist knowledge and the given background.
Based on the known existing knowledge, generate new conjectures in the following format:

(1) xxx
(2) xxx
(3) xxx

Be sure to use English answers (proper nouns need to be marked in English), statements as concise and academic as possible, do not have too much repetitive information, numerical values using the original numbers, be sure to strictly follow the format, the corresponding content output to xxx.

Note: Please respond directly to the multiple hypotheses without adding any extra sentences.
Now give hypothesis based on the following background:
{user_input}

---

Table 10: Prompt for zero-shot hypothesis generation.

### F.2    Prompts for Multi-agent Collaboration

We present prompts for each role in multi-agent collaboration in Table 12, and prompts for environment settings in Table 13.

**FEW-SHOT EXAMPLES FOR HYPOTHESIS GENERATION.**
You are a renowned biomedical researcher. You can give novel hypothesis for the background based on your exist knowledge. Please follow the given examples and give the hypothesis in the SINGLE TURN.

Background:
(1) Neonatal intensive care is associated with long-term health problems in children such as cerebral palsy, mental retardation, deafness, blindness, learning disabilities, and behavioral problems.
(2) Mothers of preterm infants experience more severe psychological distress compared to mothers of healthy full-term infants, but the impact of caregiving on parents of children discharged from NICUs is not well-researched.
(3) Parents of NICU children show no difference in psychosocial health compared to parents of healthy full-term children.
Hypothesis:
(1) The mental health of parents of NICU children may improve over time due to adaptation and relief from initial fear and anxiety.
(2) Child characteristics, such as health status, behavior problems, and birth-related risk factors, may influence parental psychosocial health.
(3) Certain factors, such as caregiver strain, family function, and demographic variables, may predict parental psychosocial health.

Background:
(1) Recruitment of tumor supporting stromal cells and tissue remodeling in the tumor microenvironment support cancer cell proliferation, invasion, metastasis, and drug resistance.
(2) Mesenchymal stem cells (MSC) are recruited by cancer cells into the tumor site and play a role in modulating tumor progression.
(3) Intratumoral heterogeneity exists in solid tumors, with cancer stem cells (CSCs) and clonal evolution contributing to the complexity of cancer. Hypothesis:
(1) Transcriptional regulators are responsible for tumor-supporting stromal reprogramming, specifically in MSC in the tumor stroma.
(2) Intercellular communication between cancer cells and recruited MSCs is mediated by cell-to-cell contact, paracrine interactions, and microvesicles.
(3) Epithelial cancer cell plasticity is regulated by tumor stroma interaction signals, enabling non-CSCs to convert into CSCs.

...
Background: {input}
Hypothesis:

Table 11: Manually constructed context examples of background-hypothesis pairs sampling from literatures before January 2023.

---

### PROMPTS FOR ROLE DESIGN IN MULTI-AGENT COLLABORATION

**Analyst**:
You are the Analyst. Depending on the phase of the iteration, your role may slightly differ:

- **Initial Phase**: Analyze the provided research background to distill its core components into pivotal keywords or topics. This will set the stage for the Engineer's search efforts.
- **Feedback Phase**: Based on feedback from the Critic, you might need to re-analyze the research background or provide additional insights to refine the search direction.

In either case, ensure clarity and relevance in your analysis. Conclude by listing the identified keywords or topics or by providing revised insights.

**Engineer**:
You are the Engineer. Your task revolves around searching based on the received keywords or insights, and this can involve multiple iterations:

- Plan your search strategies by crafting logical keyword combinations.
- Conduct systematic searches for each combination, meticulously gathering data and results.
- Refine your searches iteratively based on initial findings and any new insights from the Analyst.

Your output should be comprehensive and organized. For each keyword combination:

- **Title of Source**: Provide the title of the paper, article, or material you've found.
- **Abstract/Summary**: A brief summary or the abstract of the source.
- **Key Findings**: Highlight pivotal points or findings from the source that are relevant to the research background.
- **Implications**: If any, mention the implications or significance of the findings.
- **Relevant Quotes/Excerpts**: Extract direct quotes or sections that are particularly insightful.
Group your findings into individual "clues" based on themes or topics that emerge. This structure will provide the Scientist with detailed and organized data, enabling them to craft a robust hypothesis.

Conclude by presenting the structured "clues" for each keyword combination.

**Scientist**:
You are the Scientist. Your task is to craft a hypothesis based on the Engineer's findings and the initial research background:

- Derive a potential hypothesis that bridges the existing literature with new insights.
- Ensure the hypothesis is both innovative and scientifically grounded.

Clearly state the proposed hypothesis, preparing it for evaluation by the Critic.

**Critic**:
You are the Critic, responsible for evaluating the collaborative endeavor. Scrutinize the Scientist's hypothesis in light of the 'Research Background'. Gauge its novelty, coherence, and scientific validity. Should the hypothesis necessitate refinement:

- Clearly articulate feedback, specifying areas needing improvement.
- Instruct the Analyst to either re-evaluate the 'Research Background' or offer new insights to reshape the Engineer's subsequent search iteration.

When the hypothesis aligns with expectations and meets the desired standards, present and approve it using the structured format:

Final Answer:
(1) [First Point or Aspect of the Hypothesis]
(2) [Second Point or Aspect of the Hypothesis]
(3) [Third Point or Aspect of the Hypothesis]
...

---

Table 12: Prompts for role design in multi-agent collaboration on hypothesis proposing task.

---

**PROMPT FOR ENVIRONMENT SETTING IN MULTI-AGENT COLLABORATION.**

Topic Prompt for All Agents:

You are part of a collaborative multi-agent system designed to propose a hypothesis based on a given research background. Each of you has a specific role:

- **Analyst**: Analyzes the research background, distills its essence, and provides pivotal keywords or topics for further exploration.
- **Engineer**: Uses the keywords to plan and conduct systematic searches, meticulously gathering and organizing findings into detailed and structured "clues".
- **Scientist**: Crafts a potential hypothesis based on the organized findings and the original research background.
- **Critic**: Evaluates the hypothesis for its novelty, coherence, and scientific validity, providing feedback for refinement if necessary.

Your collaboration is iterative. Based on feedback from the Critic, the process can loop back to the Analyst for refined insights, leading to new searches by the Engineer, and a refined hypothesis by the Scientist.

Stay focused on your individual roles, collaborate effectively, and aim to derive a well-informed, novel hypothesis based on the research background provided.

Research Background:
background

Objective:
Using the research background and collaborative insights, the goal is to construct the most logical and scientifically robust hypothesis. Let's collaborate effectively to achieve this.

---

Table 13: Prompts for environment setting in multi-agent collaboration.

---

**PROMPT FOR CHATGPT EVALUATION ON NOVELTY METRIC.**

You are an expert in biomedicine.
Evaluate the novelty of the generated scientific hypothesis and the given background.
The score range should be 0 to 3. 0 means there's no novelty, which indicates that the hypothesis is a paraphrase of the background. 1 means there's slight novelty. 2 means there's moderate novelty. 3 means the hypothesis has strong novelty, which gives new insights beyond the background. Output is an integer.

Please provide a step-by-step explanation supporting your score.
At the end of your response, clearly state the score in the format 'Score: [value]', where [value] can be 1, 2, or 3.

Background: {background}
Generated scientific hypothesis: {hypothesis}

---

Table 14: Prompts for ChatGPT evaluation on novelty metric.

## F.3 Prompts for ChatGPT Evaluation

The instruction formats for prompting ChatGPT for evaluation on novelty, relevance, significance, and verifiability are displayed in Table 14, Table 15, Table 16, and Table 17, respectively.

## F.4 Finetuning Implementation

The fine-tuning process consists of three epochs, employing a batch size of 8, a maximum sequence length of 2048 tokens, and a learning rate set at 3e-5. We implement early stopping and select the best checkpoints based on their performance on the seen test dataset.

---

**PROMPT FOR CHATGPT EVALUATION ON RELEVANCE METRIC.**

You are an expert in biomedicine.
Evaluate the relevance of the generated scientific hypothesis and the given background.
The score range should be 0 to 3. 0 means there's no relevance. 1 means there's slight relevance. 2 means there's moderate relevance. 3 means they are strongly related. Output is an integer.

Please provide a step-by-step explanation supporting your score.
At the end of your response, clearly state the score in the format 'Score: [value]', where [value] can be 1, 2, or 3.

Background: {background}
Generated scientific hypothesis: {hypothesis}

---

Table 15: Prompts for ChatGPT evaluation on relevance metric.

---

**PROMPT FOR CHATGPT EVALUATION ON SIGNIFICANCE METRIC.**

You are an expert in biomedicine.
Evaluate the significance of the generated scientific hypothesis and the given background.
The score range should be 0 to 3. 0 means there's no significance, which indicates that the hypothesis is just a common knowledge. 1 means there's slight significance. 2 means there's moderate significance. 3 means the hypothesis has strong significance, which gives significant insights beyond the background. Output is an integer.

Please provide a step-by-step explanation supporting your score.
At the end of your response, clearly state the score in the format 'Score: [value]', where [value] can be 1, 2, or 3.

Background: {background}
Generated scientific hypothesis: {hypothesis}

---

Table 16: Prompts for ChatGPT evaluation on significance metric.

---

**PROMPT FOR CHATGPT EVALUATION ON VERIFIABILITY METRIC.**

You are an expert in biomedicine.
Evaluate the verifiability of the generated scientific hypothesis and the given background.
The score range should be 0 to 3. 0 means there's no verifiability, which indicates that the hypothesis is not possible to be verified in future work. 1 means there's slight verifiability. 2 means there's moderate verifiability. 3 means the hypothesis has strong verifiability, which means the hypothesis is very likely to be verified in future work. Output is an integer.

Please provide a step-by-step explanation supporting your score.
At the end of your response, clearly state the score in the format 'Score: [value]', where [value] can be 1, 2, or 3.

Background: {background}
Generated scientific hypothesis: {hypothesis}

---

Table 17: Prompts for ChatGPT evaluation on verifiability metric.

