# OpenReview forum: "Large Language Models as Biomedical Hypothesis Generators: A Comprehensive Evaluation"
_colmweb.org/COLM/2024/Conference — COLM_

### Official Review · Reviewer_uQqT · 2024-05-10

**Rating:** 7
**Confidence:** 4
**Ethics Flag:** 1

**Summary:**

The paper evaluates the large language model's ability as a biomedical hypothesis generator. It starts by constructing background hypothesis pairs from biomedical literature and splitting the test sets based on publication date to mitigate data leakage. The paper then assesses the hypothesis generation capabilities of the state-of-the-art models in the zero-shot, few-shot, and fine-tuning settings. It conducts both automatic and human evaluations. The experiment shows several impacts of external knowledge. The paper also builds a multi-agent framework for hypothesis generation and refinement. Specifically, the paper constructs four roles including analyst, engineer, scientist, and critic. The paper also tests tool learning.

**Questions To Authors:**

1. What are the sources of the dataset?
2. What are the quality of the dataset?

**Reasons To Accept:**

1. The paper proposes an interesting and useful task for generating biomedical hypotheses. To avoid data leakage, it also constructs a new dataset with 2,700 seen and 200 unseen subsets. The paper also proposes a new multi-agent hypothesis generation framework for analyzing human scientific collaboration. The multi-agent framework design is intuitive.
2. The paper conducts a comprehensive evaluation. When analyzing LLMs' ability to generate hypotheses, it mainly compares three types of state-of-the-art models: API-based LLMs, open-source LLMs, and medical domain LLMs. The paper tests those models under zero-shot, few-shot, and fine-tuning. It analyzes the generation results with traditional metrics and ChatGPT and also conducts human evaluations. The paper reports human evaluation interannotator coefficient and its correlation with ChatGPT.
3. The visualizations of the results are clear and straightforward. The paper provides additional generation results and experiment details in the appendix. The paper also provides some interesting experiments insights about external knowledge.

**Reasons To Reject:**

1. Some details of the paper are unclear. It seems that the assumption and the background pairs are generated by GPT4. While the authors admit that manual annotation of a subset could help reflect the quality of the dataset, the paper fails to provide any manual evaluation of the constructed data. Additionally, the dataset seems to be a little bit small(with 200 pairs for testing). What does the 5shot* mean? Does it mean similarity retrieval-based few-shot? The fonts in Figure 5 are too small. The title of section 3 needs to be updated since it also analyzes few-shot and fine-tuning settings.
2. While the paper also evaluates the performance of multi-agent and tool use, the analysis part is superficial compared to section 3. The paper claims that adding a tool can further enhance the multi-agent. However, its performance on relevance and verifiability dropped. Additionally, the average score is also worse.
3. The external knowledge part focuses on the few-shot part. It might be more useful to include some external knowledge graph in the experiment to further strengthen this section's analysis.

---

> ### Author Rebuttal · Authors · 2024-05-30
>
> Thank you for the time and effort you've invested in reviewing our paper. We value your feedback and will address each of your concerns as outlined below.
>
> ### Q1: Additional Details of the Paper
> We appreciate the detailed suggestions from the reviewers. For this study, we gathered real literature data from PubMed, focusing primarily on abstracts. Using GPT-4, we extracted background information and conclusions, which were then organized into hypotheses and structured data formats. This process involved collaboration with medical and biological experts, and we assessed the effectiveness of our dataset by randomly sampling and analyzing 5% of the data. Additionally, we experimented with few-shot settings, utilizing both a default 5-shot with random selection and a 5-shot* with dense retrieval to better understand the uncertainty in the background information.
>
> ### Q2: Discussing Tool Performance in a Multi-agent Framework
> The tools used in this study primarily focus on deterministic content retrieval and enhancement, which can limit the model's ability to explore and incorporate uncertainty, subsequently influencing knowledge discovery outcomes. Future work will explore the inclusion of more dynamic tools, such as multi-tools collaboration, to enhance LLMs performance. We aim to optimize strategies that introduce uncertainty via reinforcement learning to significantly improve the hypothesis generation capabilities of our models.
>
> ### Q3: Integration of External Knowledge Graphs
> Your suggestion to integrate external knowledge graphs is excellent. Currently, our approach mainly utilizes internal knowledge within the model, with the few-shot setting guiding its reasoning. Moving forward, we plan to incorporate knowledge graphs to aid in generating higher-quality hypotheses.
>
> ### Q4: Sources of the Dataset
> The dataset was constructed using high-quality collaborative filtering from PubMed. Selection was guided by recommendations from our co-authors in the field of biomedicine, ensuring that we sourced information from reputable journals.
>
> ### Q5: Quality of the Dataset
> This project involves collaboration with medical and biological experts who conduct thorough evaluations and analyses to ensure the dataset's feasibility and usability. Additionally, we perform secondary verification and assessment using GPT-4 to maintain the quality and reliability of the data.

---

> > ### Comment · Reviewer_uQqT · 2024-06-03
> > **Response to authors' rebuttal**
> >
> > Thanks for your clarification! You have addressed all of my concerns. I have read it carefully and decided to keep my positive score.

---

### Official Review · Reviewer_gBf1 · 2024-05-11

**Rating:** 8
**Confidence:** 4
**Ethics Flag:** 1

**Summary:**

The authors present a comprehensive evaluation of large language models (LLMs) as biomedical hypothesis generators. By constructing a novel temporal dataset of background-hypothesis pairs and employing innovative experimental setups, they provide valuable insights into the capabilities of LLMs in zero-shot and few-shot settings. The authors also propose a tool-augmented/multi-agent framework and multidimensional evaluation metrics, which seem to contribute to a deeper understanding of the factors influencing LLM-based hypothesis generation. Additionally, they explore the role of uncertainty and demonstrate that increasing uncertainty through tool use and multi-agent interactions can enhance the diversity and quality of generated hypotheses.

**Questions To Authors:**

1. **Protocol and standards for human evaluation**: The authors conduct human evaluations to assess the coherence of hypotheses generated by LLMs and compare them with GPT-4 evaluation scores. However, the paper would benefit from providing more details regarding the protocol and standards followed during the human evaluation process. This additional information would help readers better understand the methodology and ensure the reproducibility of the human evaluation results.

2. **Analysis of the case study**: The case study section presents a list of tables containing generated hypotheses from various models. While this information is valuable, the lack of detailed analysis accompanying the tables may confuse readers. To improve clarity and enhance the readers' understanding, the authors should provide a more thorough analysis of the case study results.

**Reasons To Accept:**

1. **Comprehensive evaluation of LLMs in zero-shot and few-shot hypothesis generation**:
The authors create a novel temporal biomedical instruction dataset and conduct innovative experiments to thoroughly analyze and evaluate the performance of LLMs in generating hypotheses.
The findings reveal that LLMs exhibit foundational higher-order reasoning capabilities and can generate novel hypotheses, offering fresh empirical insights for knowledge discovery in the biomedical domain.

2. **Uncertainty on hypothesis generation**:
The authors explore the role of uncertainty in hypothesis generation by incorporating tool use and multi-agent interactions. They find that increasing uncertainty through these methods can facilitate diverse candidate generation and improve zero-shot hypothesis generation performance.

3. **Contribution in the biomedical domain**:
The paper conducts a thorough evaluation of LLMs, specifically in the context of biomedical hypothesis generation, providing valuable insights and a foundation for future research in this domain.

**Reasons To Reject:**

1. **Design of evaluation metrics for evaluation without ground truth**: The paper lacks sufficient details and validation regarding the design of the four evaluation metrics.
2. **Tool use**: The authors directly employ the tool sets from ReAct and OpenAI to conclude that tool use does not significantly help with hypothesis generation. However, it is expected that general-domain tools would provide limited contributions.
3. **Role of the multi-agent framework**: It may lack ablation studies to evaluate the individual contributions and necessity of each role within the multi-agent framework.

---

> ### Author Rebuttal · Authors · 2024-05-30
>
> Thank you for your positive feedback and valuable suggestions. We will revise our paper with these comments.
>
> ### Q1: Detailing the Evaluation Metrics
> We adopt the metrics from prior research [1], including
>
> - Novelty: Does the hypothesis introduce new information or perspectives?
> - Relevance: How closely is the hypothesis related to the topic or question?
> - Significance: What impact does the hypothesis have on understanding or addressing the problem?
> - Verifiability: Can the hypothesis be tested using existing methods or data?
>
> [1]  Goal-driven discovery of distributional differences via language descriptions. NeurIPS 2023
>
> ### Q2: Exploring Limited Contributions to Tool Use
> The primary aim of this paper is to assess the hypothesis-generating capabilities of LLMs. Our current focus is on the hypothesis generation stage, using tools such as Web search and PubMed. Introducing more specialized tools into the knowledge discovery cycle is indeed valuable. We plan to integrate more bioinformatics analysis tools to enhance our research.
>
> ### Q3: Considering Ablation in a Multi-agent Framework
> Due to cost constraints, we model this framework based on the natural scientific research process, which aligns with real-world scientific activities. We plan to address this ablation and explore improved variants in future research.
>
> ### Q4: Protocols and Standards for Human Evaluation
> We establish four comprehensive evaluation metrics in #Q2. Evaluations will be conducted by a team of three experts from our cooperative institutions, each with degrees and at least three years of research experience in biomedicine. The experts could utilize databases and academic search engines to validate hypotheses.
>
> ### Q5: Enhancing Case Study Analysis
> We will expand on the case study from a biomedical perspective in Appendix E.1.
> The case involves MI predictive biomarkers and two-stage power-law rheology. GPT-3.5 highlights a "fundamental biomechanical principle," Llama-2-70b-chat discusses "changes in the levels of collagen and proteoglycans in the extracellular matrix," and WizardLM-70B-V1.0 considers "the progression of myocardial damage and remodeling." These models propose hypotheses that align well with the relevant background and offer novel insights, demonstrating originality and more than mere information extraction. Conversely, PMC-LLaMA-13B and WizardLM-13B-V1.2 provided only straightforward interpretations of the results without introducing new content.

---

> > ### Comment · Reviewer_gBf1 · 2024-06-01
> > **Thank you for your response**
> >
> > I acknowledge that I have read the responses by the authors and I will keep my positive score.

---

### Official Review · Reviewer_hFVW · 2024-05-13

**Rating:** 7
**Confidence:** 4
**Ethics Flag:** 1

**Summary:**

This paper proposed a novel dataset with rigorous curation, as well as some interesting/new metrics to evaluate/test the quality of the hypothesis of biofield. The authors used both human assessment and GPT-4 assessment in their evaluation. Many interesting discoveries were revealed and discussed, which will inform the community in the long term.

**Questions To Authors:**

1. Most of the baselines of Table 1 focus on not most advanced models. There are many other state of the art, for example, Mistral MOE models.  Is there a reason for this choice?

**Reasons To Accept:**

1. Authors curate a dataset of background hypothesis pairs from biomedical papers and classify them into training, seen, and unseen partitions based on their publication date to avoid the papers potentially included in LLM's pretraining. This will help test the generalization ability of LLMs. This dataset will inform the community in the long term.

2. There are many baselines studied. Various baselines are based on different LLMs as well as different ICL methods, from agent-based baselines to tool-based baselines.

3. One of the interesting discoveries about the alignment of AI feedback and human assessment shows a strong correlation, signaling that incorporating LLMs into hypothesis generation is possible.

**Reasons To Reject:**

1. Dataset validation and cleaning. Based on Figure 1, it seems that most of the curation is made by GPT-4, and human intervention is not very much involved. This might somehow bring bias and it might be useful to have human annotators to evaluate the error and bias rate.

2. Minor point: qualitative studies might be included to have a better sense of the alignment of human assessment and LLMs, in the cases where they agree and they disagree.

---

> ### Author Rebuttal · Authors · 2024-05-30
>
> Thank you for your thoughtful feedback and comments. We appreciate your enthusiasm and insights. Below, we provide detailed responses to your questions.
>
> ### Q1: Dataset Validation and Cleaning by GPT-4 and Experts
> We collaborated with experts in medicine and biology and adopted a 5% sampling rate for expert evaluation. Given the constraints of manpower and costs, a full-scale evaluation was not feasible. However, we believe that a 5% sampling rate provides a statistically representative overview and effectively reflects the overall data quality. We value your understanding and support in this matter.
>
> ### Q2: Incorporating Agreement for Qualitative Studies
> Initially, our goal for manual evaluation was to fine-tune the assessments made by the LLM and make necessary corrections. Considering the high-throughput capabilities of LLMs in knowledge discovery, automatic evaluation is essential for pinpointing high-quality findings. This method aligns with the strategies used in Reward Models and the Critic role in Multi-agent systems. We acknowledge that introducing qualitative studies could enhance our understanding of how well human assessments and LLM outputs align, especially regarding areas of agreement and disagreement. We plan to delve into this area in our future research.
>
> ### Q3: Evaluation of More Advanced LLMs
> During our experimental phase, we carefully selected a data split from before January 2023 and used models trained prior to this date to prevent data contamination and ensure unbiased evaluation. We recognize that newer, more powerful models hold promising potential for advancing knowledge discovery. In our forthcoming studies, we aim to utilize continuously updated test data and further investigate the capabilities of advanced models like Llama-3 and Mistral MoE in driving forward knowledge discovery.

---

> > ### Comment · Reviewer_hFVW · 2024-06-04
> > **reponse**
> >
> > thanks authors for the response, and I remain positive about the paper.

---

### Official Review · Reviewer_7hvv · 2024-05-15

**Rating:** 6
**Confidence:** 4
**Ethics Flag:** 1

**Summary:**

This work explores the use of large language models (LLMs) in generating hypotheses for biomedical research. It introduces a novel dataset specifically designed to avoid data contamination and assesses these models across various settings, including zero-shot and few-shot scenarios, using several quantitative metrics to evaluate the quality of the generated hypotheses. Methodologically, it integrates 'tool use' and multi-agent interactions to determine how these approaches can enhance the hypothesis generation capabilities of LLMs. The results indicate that LLMs can, to some extent, generate relevant hypotheses, which could potentially accelerate biomedical research and discovery.

**Reasons To Accept:**

The work introduces a relevant resources for future research, and conduct several evaluation that may serve as starting point for more detailed analyses.

**Reasons To Reject:**

Overall, the paper could benefit from a more stricter contextualisation and focus, given the challenging nature of the tasks.
- First, the work would benefit from a more detailed analysis of the types of hypotheses that LLMs can generate from a biomedical perspective, rather than just assessing the general quality of the hypotheses. This would better demonstrate the models' generalisation across various biomedical research scenarios.
- The discussion about 'dark knowledge' is vaguely defined and poorly explained, leaving readers uncertain about its relevance and impact on hypothesis generation, and the strength of the supporting evidence for such statements.
- Additionally, the discussion on uncertainty and the rationale behind certain evaluation metrics and frameworks, like SelfBLEU and the agent-based framework, is overly general and not well-supported, limiting the reliability of the evaluation process.
- Lastly, a more elaborated analysis that explicitly compares this contribution with existing literature would better contextualize and strengthen the novelty of the proposed approaches.

---

> ### Author Rebuttal · Authors · 2024-05-30
>
> Thank you for your constructive feedback and valuable suggestions.
>
> ### Q1: Types of Hypotheses from a Biomedical Perspective
>
> By integrating the classification from medical statistics, we analyze the generated hypotheses and found most are non-directional and simplistic, while only a few are directional and complex. This observation highlights significant room for LLMs to generate high-value, complex hypotheses. Given the limitations of space in this paper, detailed analyses of specific scenarios will be addressed in our future work.
>
> ### Q2: "Dark Knowledge" and Hypothesis Generation
>
> The term "dark knowledge" refers to the hidden, parameterized insights within our model, inspired by Geoffrey Hinton's concept of knowledge distillation. This model serves a dual purpose: firstly, to formulate explicit knowledge, and secondly, to foster the development of "dark knowledge" — to amalgamate existing knowledge to forge new insights. This capability is crucial as it enhances LLMs' ability to generalize and adapt to new, unknown areas, aligning with our core goals of advancing knowledge discovery.
>
> ### Q3: Fine-Grained Evaluation of Uncertainty
>
> Addressing uncertainty in LLMs' predictions is crucial. There are many methods for uncertainty computation, including entropy/semantic/consistency-based[1,2] approaches. However, these methods introduce a high computational cost, particularly due to the need for multiple sampling iterations, which makes them impractical for our experiments. Given the pioneering nature of our research in applying LLMs to knowledge discovery, and to keep the evaluation manageable within the confines of this paper, we opt to use SelfBLEU [3] to assess internal uncertainty among hypotheses. We plan to explore more methods in future research.
>
> - [1] Semantic uncertainty: Linguistic invariances for uncertainty estimation in natural language generation.
> - [2] Can llms express their uncertainty? an empirical evaluation of confidence elicitation in llms.
> - [3] Jointly measuring diversity and quality in text generation models.
>
> ### Q4: Comparative Analysis with Existing Literature
>
> Previous studies primarily focus on deriving and extrapolating from established knowledge. However, there has been a lack of thorough evaluation and analysis concerning the capacity of LLMs to generate viable hypotheses. To the best of our knowledge, our paper is the first to confirm LLMs hold the ability to formulate hypotheses. We will include these in the revision.

---

### Decision · Program_Chairs · 2024-07-10

**Decision:**

Accept

**Comment:**

This paper explores the capabilities of LLMs as (biomedical) hypothesis generators. There was consensus amongst reviewers that this i novel task which might inspire follow-up work, and the work here is conducted with reasonable rigor (namely the dataset curation and the range of models evaluated).

There are some concerns regarding the evaluation, but then this is an inherently difficult sort of thing to evaluate empirically, and this in my view should not preclude research on new (hard to evaluate) applications of LLMs.

[comments from PCs] Figures are not readable. It's critical to improve them.